# Compressive Sensing of Signals from a GMM with Sparse Precision Matrices

[1]**Jianbo Yang**    [1]**Xuejun Liao**    [2]**Minhua Chen**    [1]**Lawrence Carin**

[1]Department of Electrical and Computer Engineering, Duke University

[2]Department of Statistics & Department of Computer Science, University of Chicago

{jianbo.yang;xjliao;lcarin@duke@duke.edu},{dukemeeting@gmail.com}

## Abstract

This paper is concerned with compressive sensing of signals drawn from a Gaussian mixture model (GMM) with sparse precision matrices. Previous work has shown: (i) a signal drawn from a *given* GMM can be perfectly reconstructed from $r$ noise-free measurements if the (dominant) rank of each covariance matrix is less than $r$; (ii) a sparse Gaussian graphical model can be efficiently estimated from *fully-observed* training signals using graphical lasso. This paper addresses a problem more challenging than both (i) and (ii), by assuming that the GMM is *unknown* and each signal is only observed through incomplete linear measurements. Under these challenging assumptions, we develop a hierarchical Bayesian method to simultaneously estimate the GMM and recover the signals using solely the incomplete measurements and a Bayesian shrinkage prior that promotes sparsity of the Gaussian precision matrices. In addition, we provide theoretical performance bounds to relate the reconstruction error to the number of signals for which measurements are available, the sparsity level of precision matrices, and the "incompleteness" of measurements. The proposed method is demonstrated extensively on compressive sensing of imagery and video, and the results with simulated and hardware-acquired real measurements show significant performance improvement over state-of-the-art methods.

## 1   Introduction

Gaussian mixture models (GMMs) [1, 2, 3] have become a popular signal model for compressive sensing [4, 5] of imagery and video, partly because the information domain in these problems can be decomposed into subdomains known as pixel/voxel patches [3, 6]. A GMM employs a Gaussian precision matrix to capture the statistical relations between local pixels/voxels within a patch, and meanwhile captures the global statistics between patches using its clustering mechanism.

Compressive sensing (CS) of signals drawn from a GMM admits closed-form minimum mean squared error (MMSE) reconstruction from linear measurements. Recent theoretical analysis in [7] shows that, given a sensing matrix with entries i.i.d. drawn from a zero-mean, fixed-variance, Gaussian distribution or Bernoulli distribution with parameter 0.5, if the GMM is known and the (dominant) rank of each covariance matrix is less than $r$, each signal can be perfectly reconstructed from $r$ noise-free measurements. Though this is a much less stringent reconstruction condition than that prescribed by standard restricted-isometry-property (RIP) bounds, it relies on the assumption of knowing the exact GMM. If a sufficient number of fully observed signals are available beforehand, one can use maximum likelihood (ML) estimators to train a GMM [8, 9, 7, 1, 10] for use in reconstructing the signals in question. Unfortunately, finding an accurate GMM *a priori* is usually a challenge in practice, because it is difficult to obtain training signals that match the statistics of the interrogated signals.

Recent work [2] on GMM-based methods proposes to solve this problem by estimating the Gaussian components, based on measurements of the signals under interrogation, without resorting to any fully-observed signals to train a model in advance. The method of [2] has two drawbacks: (i) it estimates full dense Gaussian covariance matrices, with the number of free parameters to be estimated growing quadratically fast with the signal dimensionality $n$; (ii) it does not have performance guarantees, because all previous theoretical results, including those in [7], assume the GMM is given and thus are no longer applicable to the method of [2]. This paper addresses these two issues.

First, we effectively reduce the number of GMM parameters by restricting the GMM to have sparse precision matrices with group sparsity patterns, making the GMM a mixture of group-sparse Gaussian graphical models. The group sparsity is motivated by the Markov random field (MRF) property of natural images and video [11, 12, 13]. Instead of having $n^2$ parameters for each Gaussian component as in [2], we have only $n + s$ parameters, where $s$ is the number of nonzero off-diagonals of the precision matrix. We develop a variational maximum-marginal-likelihood estimator (variational MMLE) to simultaneously estimate the GMM and reconstruct the signals, with a Bayesian shrinkage prior used to promote sparsity of the Gaussian precision matrices. Our variational MMLE maximizes the marginal likelihood of the GMM given only the linear measurements, with the unknown signals treated as random variables and integrated out of the likelihood. A key step of the variational MMLE is using Bayesian graphical lasso to reestimate the sparse Gaussian precision matrices based on *a posteriori* signal samples conditional on the linear measurements.

Second, we provide theoretical performance bounds under the assumption that the GMM is not exactly known. Assuming the GMM has sparse precision matrices, our theoretical results relate the signal reconstruction error to the number of signals for which measurements are available, the sparsity level of the precision matrices, and the "incompleteness" of measurements, where the last is defined as the uncertainty (variance) of a signal given its linear measurements.

In the experiments, we present reconstruction results of the proposed method on both simulated measurements and real measurements acquired by actual hardware [6]. The proposed method outperforms the state-of-art CS reconstruction algorithms by significant margins.

**Notations**. Let $\mathcal{N}(\boldsymbol{x}|\boldsymbol{\mu}, \boldsymbol{\Omega}^{-1})$ denote a Gaussian density of $\boldsymbol{x}$ with mean $\boldsymbol{\mu}$ and precision matrix $\boldsymbol{\Omega}$, $\|\boldsymbol{M}\|_F$ denote the Frobenius matrix norm of matrix $\boldsymbol{M}$, $\|\boldsymbol{M}\|_{\max}$ denote the largest entry of $\boldsymbol{M}$ in terms of magnitude, $\mathrm{tr}(\boldsymbol{M})$ denote the trace of $\boldsymbol{M}$, $\boldsymbol{\Omega}_0 = \boldsymbol{\Sigma}_0^{-1}$ denote the true precision matrix (*i.e.*, the inverse of true covariance matrix $\boldsymbol{\Sigma}_0$), $\boldsymbol{\Omega}^*$ denote the estimate of $\boldsymbol{\Omega}_0$ by the proposed model. Herein, the eigenvalues of $\boldsymbol{\Sigma}_0$ are assumed to be bounded in a constant interval $[\tau_1, \tau_2] \subset (0, \infty)$, to guarantee the existence of $\boldsymbol{\Omega}_0$. For functions $f(x)$ and $g(x)$, we write $f(x) \asymp g(x)$ when $f(x) = \mathcal{O}(g(x))$ and $g(x) = \mathcal{O}(f(x))$ hold simultaneously.

## 2 Learning a GMM of Unknown Signals from Linear Measurements

### 2.1 Signal Reconstruction with a Given GMM

The linear measurement of an unknown signal $\boldsymbol{x} \in \mathbb{R}^n$ can be written as $\boldsymbol{y} = \boldsymbol{\Phi}\boldsymbol{x} + \boldsymbol{\epsilon}$, where $\boldsymbol{\Phi} \in \mathbb{R}^{m \times n}$ is a sensing matrix, and $\boldsymbol{\epsilon} \in \mathbb{R}^m$ denote measurement noises (we are interested in $m < n$). Assuming $\boldsymbol{\epsilon} \in \mathcal{N}(\boldsymbol{\epsilon}|\boldsymbol{0}, \mathbf{R})$, one has $p(\boldsymbol{y}|\boldsymbol{x}) = \mathcal{N}(\boldsymbol{y}|\boldsymbol{\Phi}\boldsymbol{x}, \mathbf{R})$. We further assume $\mathbf{R}$ to be a scaled identity matrix, $\mathbf{R} = \kappa^{-1}\boldsymbol{I}$, and thus the noise is white Gaussian.

If $\boldsymbol{x}$ is governed by a GMM, i.e., $p(\boldsymbol{x}) = \sum_{z=1}^{K} \pi^{(z)} \mathcal{N}(\boldsymbol{x}|\boldsymbol{\mu}^{(z)}, \boldsymbol{\Omega}^{(z)^{-1}})$, one may obtain

$$p(\boldsymbol{y}, \boldsymbol{x}, z) = \pi^{(z)} \mathcal{N}(\boldsymbol{y}|\boldsymbol{\Phi}\boldsymbol{x}, \mathbf{R}) \mathcal{N}(\boldsymbol{x}|\boldsymbol{\mu}^{(z)}, \boldsymbol{\Omega}^{(z)^{-1}}),$$

$$p(\boldsymbol{y}) = \sum_{z=1}^{K} \pi^{(z)} \mathcal{N}(\boldsymbol{y}|\boldsymbol{\Phi}\boldsymbol{\mu}^{(z)}, \mathbf{R} + \boldsymbol{\Phi}\boldsymbol{\Omega}^{(z)^{-1}}\boldsymbol{\Phi}'), \qquad p(\boldsymbol{x}, z|\boldsymbol{y}) = \rho^{(z)} \mathcal{N}(\boldsymbol{x}|\boldsymbol{\eta}^{(z)}, (\mathbf{C}^{(z)})^{-1}), \quad (1)$$

where

$$\mathbf{C}^{(z)} = \left(\boldsymbol{\Phi}'\mathbf{R}^{-1}\boldsymbol{\Phi} + \boldsymbol{\Omega}^{(z)}\right)^{-1}, \qquad \boldsymbol{\eta}^{(z)} = \boldsymbol{\mu}_z + \mathbf{C}^{(z)}\boldsymbol{\Phi}'\mathbf{R}^{-1}(\boldsymbol{y} - \boldsymbol{\Phi}\boldsymbol{\mu}_z),$$

$$\rho^{(z)} = \frac{\pi^{(z)} \mathcal{N}(\boldsymbol{y}|\boldsymbol{\Phi}\boldsymbol{\mu}^{(z)}, \mathbf{R} + \boldsymbol{\Phi}\boldsymbol{\Omega}^{(z)^{-1}}\boldsymbol{\Phi}')}{\sum_{l=1}^{K} \pi^{(l)} \mathcal{N}(\boldsymbol{y}|\boldsymbol{\Phi}\boldsymbol{\mu}^{(l)}, \mathbf{R} + \boldsymbol{\Phi}\boldsymbol{\Omega}^{(l)^{-1}}\boldsymbol{\Phi}')}. \quad (2)$$

When the GMM is exactly known, the signal is reconstructed analytically as the conditional mean,

$$\widehat{\boldsymbol{x}} \triangleq \mathbb{E}(\boldsymbol{x}|\boldsymbol{y}) = \sum_{z=1}^{K} \rho^{(z)} \boldsymbol{\eta}^{(z)}. \quad (3)$$

It has been shown in [7] that, if the (dominant) rank of each Gaussian covariance matrix is less than $r$, the signal can be perfectly reconstructed from only $r$ measurements in the low-noise regime.

## 2.2 Restriction of the GMM to a mixture of Gaussian Markov Random Fields

A Markov random field (MRF), also known as an undirected graphical model, provides a graphical representation of the joint probability distribution over multiple random variables, by considering the conditional dependences among the variables [11, 12, 13]. In image analysis, each node of an MRF corresponds to a pixel of the image in question, and an edge between two nodes is often modeled by a potential function to characterize the conditional dependence between the associated pixels. Because of the local smoothness structure of images, the edges of an MRF are usually chosen based on a pairwise neighborhood structure: each pixel *only* has edge connections with its neighbors. The widely used scheme is that each pixel *only* has edge connections with its four immediate neighboring pixels to the left, right, top and bottom [11]. Therefore, an MRF for image representation is an undirected graph with only a limited number of edges between its nodes.

Generally, learning and inference of an MRF are nontrivial, due to the nonlinearity and nonconvexity of the potential functions [14]. A popular special case of MRF is the Gaussian Markov random field (GMRF) which is an MRF with a multivariate Gaussian distribution over node variables. The best-known advantage of a GMRF is its simplicity of learning and inference, because of the nice properties of a multivariate Gaussian distribution. According to Hammersley-Clifford's theorem [15], the conditional dependence of the node variables in a GMRF is encoded in the precision matrix. As mentioned before, an MRF is sparse for image analysis problems, on account of the neighborhood structure in the pixel domain. Therefore, the multivariate Gaussian distribution associated with a GMRF has a sparse precision matrix. This property of a GMRF in image analysis is demonstrated in Section 1 of the Supplementary Material.

Inspired by the GMRF interpretation, we place a shrinkage prior on each precision matrix to promote sparsity when estimating the GMM. The Laplacian shrinkage prior used in [16] is chosen, but other shrinkage priors [17] could also be used. Specifically, we impose a Laplacian shrinkage prior on the off-diagonal elements of each of $K$ precision matrices,

$$p(\boldsymbol{\Omega}^{(k)}) = \prod_{i=1}^{n}\prod_{j<i} \frac{\sqrt{\tau^{(k)}\gamma_{ij}^{(k)}}}{2}\exp(-\sqrt{\tau^{(k)}\gamma_{ij}^{(k)}}|\omega_{ij}^{(k)}|), \quad \forall k = 1,\ldots,K, \tag{4}$$

with the symmetry constraints $\omega_{ij}^{(k)} = \omega_{ji}^{(k)}$. In (4), $\tau^{(k)} > 0$ is a "global" scaling parameter for all the elements of $\{\omega_{ij}^{(k)}|i = 1,...,n, \ j < i\}$ and generally fixed to be one [18], and $\gamma_{ij}^{(k)}$ is a "local" weight for the element $\omega_{ij}^{(k)}$. With the Laplacian prior (4), many off-diagonal elements of $\boldsymbol{\Omega}^{(k)}$ are encouraged to be close to zero. However, in the inference procedure, the above Laplacian shrinkage prior (4) is inconvenient due to the lack of analytic updating expressions. This issue is overcome by using an equivalent scale mixture of normals representation [16] of (4) as shown below:

$$\frac{\sqrt{\tau^{(k)}\gamma_{ij}^{(k)}}}{2}\exp(-\sqrt{\tau^{(k)}\gamma_{ij}^{(k)}}|\omega_{ij}^{(k)}|) = \int \mathcal{N}(\omega_{ij}^{(k)}|0,\tau^{(k)^{-1}}\alpha_{ij}^{(k)^{-1}})\mathrm{InvGa}(\alpha_{ij}^{(k)}|1,\frac{\gamma_{ij}^{(k)}}{2})\mathrm{d}\alpha_{ij}^{(k)} \tag{5}$$

where $\alpha_{ij}^{(k)}$ is an augmented variable drawn from an inverse gamma distribution. Further, one may place a gamma prior on $\gamma_{ij}^{(k)}$. Then, a draw of the precision matrix may be represented by

$$\boldsymbol{\Omega}^{(k)} \sim \prod_{i=1}^{n}\prod_{j<i}\mathcal{N}(\omega_{ij}^{(k)}|0,\tau^{(k)^{-1}}\alpha_{ij}^{(k)^{-1}}), \quad \alpha_{ij}^{(k)} \sim \mathrm{InvGa}(\alpha_{ij}^{(k)}|1,\frac{\gamma_{ij}^{(k)}}{2}), \quad \gamma_{ij}^{(k)} \sim \mathrm{Ga}(\gamma_{ij}^{(k)}|a_0,b_0) \tag{6}$$

where $a_0,b_0$ are the hyperparameters.

Suppose $\{\boldsymbol{x}_i\}_{i=1}^{N}$ are samples drawn from $\mathcal{N}(\boldsymbol{x}|\boldsymbol{0},\boldsymbol{\Omega}^{(k)^{-1}})$ and $\boldsymbol{S}$ denotes the empirical covariance matrix $\frac{1}{N}\sum_{i=1}^{N}(\boldsymbol{x}_i - \overline{\boldsymbol{x}})(\boldsymbol{x}_i - \overline{\boldsymbol{x}})'$ where $\overline{\boldsymbol{x}}$ is the empirical mean of $\{\boldsymbol{x}_i\}_{i=1}^{N}$. If the elements $\boldsymbol{\Omega}^{(k)}$ are drawn as in (6), the logarithm of the joint likelihood can be expressed as

$$\log p(\{\boldsymbol{x}_i\}_{i=1}^{N},\boldsymbol{\Omega}^{(k)}) \propto \frac{N}{2}\left(\log\det(\boldsymbol{\Omega}^{(k)}) - \mathrm{tr}(\boldsymbol{S}\boldsymbol{\Omega}^{(k)}) - \sum_{i=1}^{n}\sum_{j<i}\frac{2}{N}\sqrt{\tau^{(k)}\gamma_{ij}^{(k)}}|\omega_{ij}^{(k)}|\right). \tag{7}$$

From the optimization perspective, the maximum *a posterior* (MAP) estimations of $\boldsymbol{\Omega}^{(k)}$ in (7) is known as the adaptive graphical lasso problem [18].

## 2.3 Group sparsity based on banding patterns

The Bayesian adaptive graphical lasso described above assumes the precision matrix is sparse, and the same Laplacian prior is imposed on all off-diagonal elements of the precision matrix without any discrimination. However, the aforementioned neighborhood structure of image pixels implies that the entries of the precision matrix corresponding to the pairs between neighboring pixels tend to have significant values. This is consistent with the observations as seen from the demonstration in Section 1 of the Supplementary Material: (i) the bands scattered along a few lines above or below the main diagonal are constituted by the entries with significant values in the precision matrix; (ii) the entries in the bands correspond to the pairwise neighborhood structure of the graph, since vectorization of an image patch is constituted by stacking all columns of pixels in a patch on the top of each other; (iii) the existence of multiple bands in some Gaussian components reveals that, besides the four immediate neighboring pixels, other indirected neighboring pixels may also lead to nonnegligible conditional dependence, though the entries in the associated bands have relatively smaller values.

Inspired by the banding patterns mentioned above, we categorize the elements in the set $\{\omega_{ij}^{(k)}\}_{i=1,j<i}^n$ into two groups $\{\omega_{ij}^{(k)}|(i,j) \in \mathcal{L}_1\}$ and $\{\omega_{ij}^{(k)}|(i,j) \in \mathcal{L}_2\}$, where $\mathcal{L}_1$ denotes the set of indices corresponding to the elements in the bands and $\mathcal{L}_2$ represents the set of indices for the elements not in the bands. For the elements in the group $\{\omega_{ij}^{(k)}|(i,j) \in \mathcal{L}_2\}$, the Laplacian prior is used to encourage a sparse precision matrix. For the elements in the group $\{\omega_{ij}^{(k)}|(i,j) \in \mathcal{L}_1\}$ , the sparsity is not desired so a normal prior with Gamma hyperparameters is used instead. Accordingly, the expressions in (6) can be replaced by

$$
\boldsymbol{\Omega}^{(k)} \sim \prod_{i=1}^n \prod_{i<j} \mathcal{N}(\omega_{ij}^{(k)}|0, \tau^{(k)^{-1}} \alpha_{ij}^{(k)^{-1}})
$$

$$
\alpha_{ij}^{(k)} \sim \begin{cases} \mathrm{Ga}(\alpha_{ij}^{(k)}|c_0, d_0), & \text{if } (i,j) \in \mathcal{L}_1 \\ \mathrm{InvGa}(\alpha_{ij}^{(k)}|1, \frac{\gamma_{ij}^{(k)}}{2}), & \gamma_{ij}^{(k)} \sim \mathrm{Ga}(\gamma_{ij}^{(k)}|a_0, b_0), & \text{if } (i,j) \in \mathcal{L}_2 \end{cases} .
\tag{8}
$$

With the prior distribution of $\boldsymbol{\Omega}^{(k)}$ in (6) replaced with that in (8), the joint log-likelihood in (7) changes to

$$
\log p(\{\boldsymbol{x}_i\}_{i=1}^N, \boldsymbol{\Omega}^{(k)})
$$

$$
\propto \frac{N}{2} \left( \log \det(\boldsymbol{\Omega}^{(k)}) - \mathrm{tr}(\boldsymbol{S}\boldsymbol{\Omega}^{(k)}) - \sum_{(i,j)\in\mathcal{L}1} \frac{2}{N} \tau^{(k)} \alpha_{ij}^{(k)} \|\omega_{ij}^{(k)}\|^2 - \sum_{(i,j)\in\mathcal{L}2} \frac{2}{N} \sqrt{\tau^{(k)} \gamma_{ij}^{(k)}} |\omega_{ij}^{(k)}| \right) .
\tag{9}
$$

To the best of our knowledge, the maximum *a posterior* (MAP) estimations of $\boldsymbol{\Omega}^{(k)}$ in (9) has not been studied in the family of graphical lasso or its variants, from the optimization perspective.

## 2.4 Hierarchical Bayesian model and inference

We consider the collective compressive sensing of the signals $\mathbf{X} = \{\boldsymbol{x}_i \in \mathbb{R}^n\}_{i=1}^N$ that are drawn from an unknown GMM. The noisy linear measurements of $\mathbf{X}$ are given by $\mathbf{Y} = \{\boldsymbol{y}_i \in \mathbb{R}^m : \boldsymbol{y}_i = \boldsymbol{\Phi}_i \boldsymbol{x}_i + \boldsymbol{\epsilon}_i\}_{i=1}^N$. We assume the sensing matrices to be signal-dependent to account for generality (i.e., $\boldsymbol{\Phi}_i$ depends on the signal index $i$).

The unification of signal reconstruction with a given GMM (presented in Section 2.1) and GM-RF learning with fully-observed training signals (presented in Section 2.2) leads to the following Bayesian model,

$$
\boldsymbol{y}_i|\boldsymbol{x}_i \sim \mathcal{N}(\boldsymbol{y}_i|\boldsymbol{\Phi}_i\boldsymbol{x}_i, \kappa^{-1}\boldsymbol{I}), \quad \boldsymbol{x}_i \sim \sum_{z=1}^K \pi^{(z)} \mathcal{N}(\boldsymbol{x}_i|\boldsymbol{\mu}^{(z)}, \boldsymbol{\Omega}^{(z)^{-1}}), \quad \kappa \sim \mathrm{Ga}(\kappa|e_0, f_0)
\tag{10}
$$

$$
\boldsymbol{\Omega}^{(k)} \sim \prod_{i=1}^n \prod_{i<j} \mathcal{N}(\omega_{ij}^{(k)}|0, \tau^{(k)^{-1}} \alpha_{ij}^{(k)^{-1}}), \quad \alpha_{ij}^{(k)} \sim \mathrm{InvGa}(\alpha_{ij}^{(k)}|1, \frac{\gamma_{ij}^{(k)}}{2}), \quad \gamma_{ij}^{(k)} \sim \mathrm{Ga}(\gamma_{ij}^{(k)}|a_0, b_0),
\tag{11}
$$

The expression in (11) could be replaced by (8) if the group sparsity is considered in the precision matrix. In addition to the precision matrices, we further add the following standard priors on the other parameters of the GMM to make the proposed model a full hierarchical Bayesian model,

$$
\boldsymbol{\mu}^{(k)} \sim \mathcal{N}(\boldsymbol{\mu}^{(k)}|\mathbf{m}_0, (\beta_0 \boldsymbol{\Omega}^{(k)})^{-1}), \quad \boldsymbol{\pi} \sim \mathrm{Dirichlet}(\pi^{(1)}, \dots, \pi^{(K)}|\mathbf{a}_0),
\tag{12}
$$

where $\mathbf{m}_0$, $\mathbf{a}_0$ and $\beta_0$ are hyperparameters.

We develop the inference procedure for the proposed Bayesian hierarchical model. Let the symbols $\mathbf{Z}, \boldsymbol{\mu}, \boldsymbol{\Omega}, \boldsymbol{\pi}, \boldsymbol{\alpha}, \boldsymbol{\gamma}$ denote the sets $\{z_i\}$, $\{\boldsymbol{\mu}^{(k)}\}$, $\{\boldsymbol{\Omega}^{(k)}\}$, $\{\pi^{(k)}\}$, $\{\boldsymbol{\alpha}^{(k)}\}$, $\{\boldsymbol{\gamma}^{(k)}\}$ respectively. The marginalized likelihood function is written as

$$\mathcal{L}(\boldsymbol{\Theta}) = \ln \int p(\mathbf{Y}, \boldsymbol{\Pi}, \boldsymbol{\Theta}) \mathrm{d}\boldsymbol{\Pi}$$

where $\boldsymbol{\Pi} \triangleq \{\mathbf{X}, \mathbf{Z}, \boldsymbol{\alpha}, \boldsymbol{\gamma}\}$ and $\boldsymbol{\Theta} \triangleq \{\boldsymbol{\mu}, \boldsymbol{\Omega}, \boldsymbol{\pi}, \kappa\}$ denote the set of the latent variables and parameters of the model, respectively. An expectation-maximization (EM) algorithm [19] could be used to find the optimal $\boldsymbol{\Theta}$ by alternating the following two steps

- E-step: Find $p(\boldsymbol{\Pi}|\mathbf{Y}, \boldsymbol{\Theta}^*)$ with $\boldsymbol{\Theta}^*$ computed at the M-step, and obtain the expected complete log-likelihood $\mathbb{E}_{\boldsymbol{\Pi}}(\ln p(\mathbf{Y}, \boldsymbol{\Pi}, \boldsymbol{\Theta}^*))$.
- M-step: Find an improved estimate of $\boldsymbol{\Theta}^*$ by maximizing the expected complete log-likelihood given at the E-step.

However, it is intractable to compute the exact posterior $p(\boldsymbol{\Pi}|\mathbf{Y}, \boldsymbol{\Theta})$ at the E step. We develop a variational inference approach to overcome the intractability. Based on the mean field theory [20], we approximate the posterior distribution $p(\boldsymbol{\Pi}|\mathbf{Y}, \boldsymbol{\Theta})$ by a proposal distribution $q(\boldsymbol{\Pi})$ that factorizes over the variables as follows

$$q(\boldsymbol{\Pi}) = q(\mathbf{X}, \mathbf{Z}, \boldsymbol{\alpha}, \boldsymbol{\gamma}) = q(\mathbf{X}, \mathbf{Z})q(\boldsymbol{\alpha})q(\boldsymbol{\gamma}). \tag{13}$$

Then, we find an optimal distribution $q(\boldsymbol{\Pi})$ that minimizes the Kullback-Leibler (KL) divergence $\mathrm{KL}(q(\boldsymbol{\Pi})||p(\boldsymbol{\Pi}|\mathbf{Y}, \boldsymbol{\Theta})) = \int q(\boldsymbol{\Pi}) \ln \frac{q(\boldsymbol{\Pi})}{p(\boldsymbol{\Pi}|\mathbf{Y}, \boldsymbol{\Theta})} \mathrm{d}\boldsymbol{\Pi}$, or equivalently, maximizes the evidence lower bound (ELBO) of the log-marginal data likelihood [21], denoted by $\mathcal{F}(q(\boldsymbol{\Pi}), \boldsymbol{\Theta})$,

$$\ln p(\mathbf{Y}, \boldsymbol{\Theta}) = \ln \int q(\boldsymbol{\Pi}) \frac{p(\mathbf{Y}, \boldsymbol{\Pi}, \boldsymbol{\Theta})}{q(\boldsymbol{\Pi})} \mathrm{d}\boldsymbol{\Pi} \geq \int q(\boldsymbol{\Pi}) \ln \frac{p(\mathbf{Y}, \boldsymbol{\Pi}, \boldsymbol{\Theta})}{q(\boldsymbol{\Pi})} \mathrm{d}\boldsymbol{\Pi} \triangleq \mathcal{F}(q(\boldsymbol{\Pi}), \boldsymbol{\Theta}) \tag{14}$$

where the inequality is held based on the Jensen's inequality.

With the above approximation, the entire algorithm becomes a variational EM algorithm and it iterates between the following VE-step and VM-step until convergence:

- VE-step: Find the optimal posterior distribution $q^*(\boldsymbol{\Pi})$ that maximizes $\mathcal{F}(q(\boldsymbol{\Pi}), \boldsymbol{\Theta}^*)$ with $\boldsymbol{\Theta}^*$ computed at the VM-step.
- VM-step: Find the optimal $\boldsymbol{\Theta}^*$ that maximizes $\mathcal{F}(q^*(\boldsymbol{\Pi}), \boldsymbol{\Theta})$ with $q^*(\boldsymbol{\Pi})$ computed at the VE-step.

The full update equations of the variational EM algorithm are given in Section 2 of the Supplementary Material.

## 3 Theoretical Analysis

The proposed hierarchical Bayesian model unifies the task of signal recovery and the task of estimating the mixture of GMRF, with a common goal of maximizing the ELBO of the log-marginal likelihood of the measurements. This section provides a theoretical analysis to further reveal the mutual influence between these two tasks (Theorem 1 and Theorem 2), and establish a theoretical performance bound (Theorem 3) to relate the reconstruction error to the number of signals being measured, the sparsity level of precision matrices, and the "incompleteness" of measurements. The proofs of these theorems are presented in Sections 3-5 of the Supplementary Material. For convenience, we consider the single Gaussian case, so the superscript $(k)$ is omitted in the sequel. We begin with the definitions and assumptions used in the theorems.

**Definition 3.1** *Let $\widetilde{\boldsymbol{x}}_i$ and $\widehat{\boldsymbol{x}}_i$ be the signals estimated from measurement $\boldsymbol{y}_i$, using the true precision matrix $\boldsymbol{\Omega}_0$ and the estimated precision matrix $\boldsymbol{\Omega}^*$ respectively, according to (3),*

$$\widehat{\boldsymbol{x}}_i = \boldsymbol{\mu} + \left(\boldsymbol{\Omega}_0 + \boldsymbol{\Phi}_i' \mathbf{R}^{-1} \boldsymbol{\Phi}_i\right)^{-1} \boldsymbol{\Phi}_i' \mathbf{R}^{-1} (\boldsymbol{y}_i - \boldsymbol{\Phi}_i \boldsymbol{\mu}) = \boldsymbol{\mu} + \mathbf{C}_i \boldsymbol{\Phi}_i' \mathbf{R}^{-1} (\boldsymbol{y}_i - \boldsymbol{\Phi}_i \boldsymbol{\mu})$$

$$\widetilde{\boldsymbol{x}}_i = \boldsymbol{\mu} + \left(\boldsymbol{\Omega}_0 + \boldsymbol{\Delta} + \boldsymbol{\Phi}_i' \mathbf{R}^{-1} \boldsymbol{\Phi}_i\right)^{-1} \boldsymbol{\Phi}_i' \mathbf{R}^{-1} (\boldsymbol{y}_i - \boldsymbol{\Phi}_i \boldsymbol{\mu}) = \boldsymbol{\mu} + \left(\mathbf{C}_i^{-1} + \boldsymbol{\Delta}\right)^{-1} \boldsymbol{\Phi}_i' \mathbf{R}^{-1} (\boldsymbol{y}_i - \boldsymbol{\Phi}_i \boldsymbol{\mu}) .$$

*Assuming $\boldsymbol{y}_i \in \mathbb{R}^r$ is noise-free and the (dominant) rank of $\boldsymbol{\Omega}_0$ is less than $r$, one obtains $\widehat{\boldsymbol{x}}_i$ as the true signal $\boldsymbol{x}_i$ [7], i.e., $\widehat{\boldsymbol{x}}_i = \boldsymbol{x}_i$. Then the reconstruction error of $\widetilde{\boldsymbol{x}}_i$ is $\|\boldsymbol{\delta}_i\|_2$, where $\boldsymbol{\delta}_i = \widetilde{\boldsymbol{x}}_i - \widehat{\boldsymbol{x}}_i$.*

**Definition 3.2** *The estimation error of $\mathbf{\Omega}^*$ is defined as $\|\mathbf{\Delta}\|_F$ where $\mathbf{\Delta} = \mathbf{\Omega}^* - \mathbf{\Omega}_0$.*

At each VM-step of the variational EM algorithm developed in Section 2.4, $\mathbf{\Omega}^*$ is updated based on the empirical covariance matrix $\mathbf{\Sigma}_{em}$ computed from $\{\widetilde{\boldsymbol{x}}_i\}$, i.e.,

$$\mathbf{\Sigma}_{em} = \frac{1}{N}\sum_{i=1}^{N}\widetilde{\boldsymbol{x}}_i\widetilde{\boldsymbol{x}}_i' + \frac{1}{N}\sum_{i=1}^{N}\mathbf{C}_i = \underbrace{\frac{1}{N}\sum_{i=1}^{N}\widehat{\boldsymbol{x}}_i\widehat{\boldsymbol{x}}_i'}_{\mathbf{\Sigma}_{em}^0} + \underbrace{\frac{1}{N}\sum_{i=1}^{N}(2\widehat{\boldsymbol{x}}_i\boldsymbol{\delta}_i' + \boldsymbol{\delta}_i\boldsymbol{\delta}_i' + \mathbf{C}_i)}_{\mathbf{\Sigma}_{de}}, \qquad (15)$$

where $\{\widehat{\boldsymbol{x}}_i\}$ and $\{\widetilde{\boldsymbol{x}}_i\}$ are considered to both have zero mean, as one can always center the signals with respect to their means [2].

**Definition 3.3** *The deviation of empirical matrix $\mathbf{\Sigma}_{em}^0$ is defined as $\mathbf{\Sigma}_{de} = \mathbf{\Sigma}_{em} - \mathbf{\Sigma}_{em}^0$ according to (15), and we use $\bar{\sigma}_{de} \triangleq \|\mathbf{\Sigma}_{de}\|_{\max}$ to measure this deviation. Considering the developed variational EM algorithm can converge to a local minimum, we assume $\bar{\sigma}_{de} \leq c\sqrt{\frac{\log n}{N}}$ for a constant $c > 0$[1].*

### 3.1 Theoretical results

**Theorem 1** *Assuming $\|\mathbf{C}_i\|_F\|\mathbf{\Delta}\|_F < 1$, the reconstruction error of the $i$-th signal is upper bounded as $\|\boldsymbol{\delta}_i\|_2 \leq \frac{\|\mathbf{C}_i\|_F\|\mathbf{\Delta}\|_F}{1 - \|\mathbf{C}_i\|_F\|\mathbf{\Delta}\|_F}\|\widehat{\boldsymbol{x}}_i\|_2$.*

Theorem 1 establishes the error bound of signal recovery in terms of $\mathbf{\Delta}$. In this theorem, $\mathbf{\Omega}^*$ can be obtained by any GMRF estimation methods, including [1, 2] and the proposed method.

Let $\underline{\eta} = \min_{(i,j)\in S^c}\frac{\sqrt{\tau\gamma_{ij}}}{N}, \overline{\eta} = \max_{(i,j)\in S}\frac{\sqrt{\tau\gamma_{ij}}}{N}$, $S = \{(i,j) : \omega_{ij} \neq 0, i \neq j\}$, $S^c = \{(i,j) : \omega_{ij} = 0, i \neq j\}$ and the cardinality of $S$ be $s$. The following theorem establishes an upper bound of $\|\mathbf{\Delta}\|_F$ on account of $\mathbf{\Sigma}_{de}$.

**Theorem 2** *Given the empirical covariance matrix $\mathbf{\Sigma}_{em}$, if $\underline{\eta}, \overline{\eta} \asymp \sqrt{\frac{\log n}{N}} + \bar{\sigma}_{de}$, then we have $\|\mathbf{\Delta}\|_F = \mathcal{O}_p\{\sqrt{(n+s)\log n/N} + \sqrt{n+s}\bar{\sigma}_{de}\}$.*

Note that the standard graphical lasso and its variants [18, 23] assume the true signal samples $\{\mathbf{x}_i\}$ are fully observed when estimating $\mathbf{\Omega}^*$, so they correspond to the simple case that $\bar{\sigma}_{de} = 0$. Loh and Wainwright [22, Corollary 5] also provides an upper bound of $\|\mathbf{\Delta}\|_F$ taking $\mathbf{\Sigma}_{de}$ into account. However, they assume $\mathbf{\Sigma}_{em}^0$ is attainable and the proof of their corollary relies on their proposed GMRF estimation algorithm, so the theoretical result in [22] cannot be used here.

Let $\epsilon_0 = \frac{1}{N}\sum_{i=1}^{N}\|\widehat{\boldsymbol{x}}_i - \boldsymbol{\mu}\|_2$, $\upsilon = \frac{1}{N}\sum_{i=1}^{N}\text{tr}(\mathbf{C}_i)$, $\delta_{\max} = \sup_i\|\boldsymbol{\delta}_i\|_2$, $\widehat{\boldsymbol{x}}_{\max} = \sup_i\|\widehat{\boldsymbol{x}}_i\|_2$ and $\xi = \max_i\|\mathbf{C}_i\|_F$. A combination of Theorem 1 and 2 leads to the following theorem which relates the error bound of signal reconstruction to the number of partially-observed signals (observed through incomplete linear measurements), the sparsity level of precision matrices, and the uncertainty of signal reconstruction (i.e., $\upsilon$ and $\xi$) which represent the "incompleteness" of the measurements.

**Theorem 3** *Given the empirical covariance matrix $\mathbf{\Sigma}_{em}$, if $\underline{\eta}, \overline{\eta} \asymp \sqrt{\frac{\log n}{N}} + \bar{\sigma}_{de}$, $\xi\|\mathbf{\Delta}\|_F < \zeta$ where $\zeta$ is a constant and $(1 - \zeta)/\sqrt{n+s} > M\epsilon_0(\delta_{\max} + 2\widehat{\boldsymbol{x}}_{\max})\xi$ with $M$ being an appropriate constant to make $\|\mathbf{\Delta}\|_F \leq M\sqrt{(n+s)\log n/N} + M\sqrt{n+s}\bar{\sigma}_{de}$ hold with high probability, then we obtain that $\frac{1}{N}\sum_{i=1}^{N}\|\widetilde{\boldsymbol{x}}_i - \widehat{\boldsymbol{x}}_i\|_2 \leq \frac{\sqrt{(\log n)/N} + \upsilon}{(1-\zeta)/\sqrt{n+s} - M\epsilon_0(\delta_{\max} + 2\widehat{\boldsymbol{x}}_{\max})\xi}M\epsilon_0\xi$.*

From Theorem 3, we find that when the number of partially-observed signals $N$ tends to infinity and the uncertainty of signal reconstruction $\text{tr}(\mathbf{C}_i)$ tends to zero $\forall i$, the average reconstruction error $\frac{1}{N}\sum_{i=1}^{N}\|\widetilde{\boldsymbol{x}}_i - \widehat{\boldsymbol{x}}_i\|_2$ is close to zero with high probability.

## 4 Experiments

The performance of the proposed methods is evaluated on the problems of compressive sensing (CS) of imagery and high-speed video[2]. For convenience, the proposed method is termed as *Sparse-GMM* when using the non-group sparsity described in Section 2.2,

and is termed *Sparse-GMM(G)* when using the group sparsity described in Section 2.3. For Sparse-GMM(G), we construct the two groups $\mathcal{L}_1$ and $\mathcal{L}_2$ as follows : $\mathcal{L}_1 = \{(i,j) : \text{pixel } i \text{ is one of four immediate neighbors, in the spatial domain, of pixel } j, i \neq j\}$ and $\mathcal{L}_2 = \{(i,j) : i,j = 1,2,\cdots,n, i \neq j\} \setminus \mathcal{L}_1$. The proposed methods are compared with state-of-the-art methods, including: a GMM pre-trained from training patches (GMM-TP) [7, 8], a piecewise linear estimator (PLE) [2], generalized alternating projection (GAP) [24], Two-step Iterative Shrinkage/Thresholding (TwIST) [25], KSVD-OMP [26].

For the proposed methods, the hyperparameters of the scaled mixture of Gaussians are set as $\sqrt{a_0/b_0}/N \approx 300$, $c_0 = d_0 = 10^{-6}$, the hyperparameter of Dirichlet prior $\boldsymbol{\alpha}_0$ is set as a vector with all elements being one, the hyperparameters of the mean of each Gaussian component are set as $\beta_0 = 1$, and $\mathbf{m}_0$ is set to the mean of the initialization of $\{\widehat{\boldsymbol{x}}_i\}_{i=1}^N$. We fixed $\kappa = 10^{-6}$ for the proposed methods, GMM-TP and PLE. The number of dictionary elements in KSVD is set to the best in $\{64, 128, 256, 512\}$. The TwIST adopts the total-variation (TV) norm, and the results of TwIST reported here represented the best among the different settings of regularization parameter in the range of $[10^{-4}, 1]$. In GAP, the spatial transform is chosen between DCT and waveletes and the one with the best result is reported, and the temporal transform for video is fixed to be DCT.

## 4.1 Simulated measurements

**Compressive sensing of still images**. Following the single pixel camera [27], an image $\boldsymbol{x}_i$ is projected onto the rows of a random sensing matrix $\boldsymbol{\Phi}_i \in \mathbb{R}^{m \times n}$ to obtain the compressive measurements $\boldsymbol{y}_i$ for $i = 1, \ldots, N$. Each sensing matrix $\boldsymbol{\Phi}_i$ is constituted by the elements drawn from a uniform distribution in $[0,1]$. The USPS handwritten digits dataset [3] and the face dataset [28] are used in this experiment. In each dataset, we randomly select 300 images and each image is resized to the scale of $12 \times 12$. Eight settings of CS ratios are adopted with $\frac{m}{n} \in \{0.05, 0.10, 0.15, 0.20, 0.25, 0.30, 0.35, 0.40\}$. Since signal $\boldsymbol{x}_i$ in the single pixel camera represents an entire image which generally has unique statistics, it is infeasible to find suitable training data in practice. Therefore, GMM-TP and KSVD-OMP are not compared to in this experiment[4]. For PLE, Sparse-GMM and Sparse-GMM(G), the minimum-norm estimates from the measurements, $\widehat{\boldsymbol{x}}_i = \arg\min_{\boldsymbol{x}}\{\|\boldsymbol{x}\|_2^2 : \boldsymbol{\Phi}_i\boldsymbol{x} = \boldsymbol{y}_i\} = \boldsymbol{\Phi}_i'(\boldsymbol{\Phi}_i\boldsymbol{\Phi}_i')^{-1}\boldsymbol{y}_i$, $i = 1, \ldots, N$, are used to initialize the GMM. The number of GMM components $K$ in PLE, Sparse-GMM, and Sparse-GMM(G) is tuned among $2 \sim 10$ based on Bayesian information criterion (BIC).

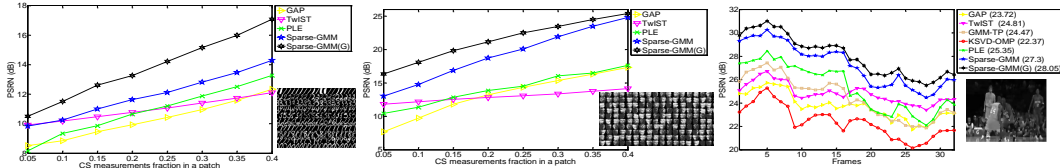

Figure 1: A comparison of reconstruction performances, in terms of PSNR, among different methods for CS of imagery on USPS handwritten digits (left) and face datasets (middle), and CS of video on NBA game dataset (right), with the average PSNR over frames shown in the brackets.

**Compressive sensing of high-speed video**. Following the Coded Aperture Compressive Temporal Imaging (CACTI) system [6], each frame of video to be reconstructed is encoded with a shifted binary mask which is designed by randomly drawing values from $\{0,1\}$ at every pixel location, with a 0.5 probability of drawing 1. Each signal $\boldsymbol{x}_i$ represents the vectorization of $T$ consecutive spatial frames, obtained by first vectorizing each frame into a column and then stacking the resulting $T$ columns on top of each other. The measurement $\boldsymbol{y}_i$ is constituted by $\boldsymbol{y}_i = \boldsymbol{\Phi}_i\boldsymbol{x}_i$ where $\boldsymbol{\Phi}_i = [\boldsymbol{\Phi}_{i,1}, \ldots, \boldsymbol{\Phi}_{i,T}]$ and $\boldsymbol{\Phi}_{i,t}$ is a diagonal matrix with its diagonal being the mask that is applied to the $t$-th frame. A video containing NBA game scenes is used in the experiment. It has 32 frames, each of size $256 \times 256$, and $T$ is set to be 8. For GMM-TP, KSVD-OMP, PLE, Sparse-GMM and Sparse-GMM(G), we partition each $256 \times 256$ measurement frame into a set of $64 \times 64$ blocks, and each block is considered as if it were a small frame and is processed independently of other blocks.[5] The patch is of size $4 \times 4 \times T$. Since each block is only $64 \times 64$, a small number of GMM components are sufficient to capture its statistics, and we find the results are robust to $K$ as long as $2 \leq K \leq 5$ for PLE, Sparse-GMM and Sparse-GMM(G). Following [8, 26], we use the patches

of a randomly-selected video containing traffic scenes[6], which are irrelevant to the NBA game, as training data to learn a GMM for GMM-TP with 20 components, and we use it to initialize PLE, Sparse-GMM, and Sparse-GMM(G). The same training data are used to learn the dictionaries for KSVD-OMP.



**Results**. From the results shown in Figure 1, we observe that the proposed methods, especially Sparse-GMM(G), outperforms other methods with significant margins in all considered settings. The better performance of Sparse-GMM(G) over Sparse-GMM validates the advantage of considering group sparsity in the model. Figure 2 shows the an example precision matrix of one of $K$ Gaussian components that are learned

Figure 2: Plots of an example precision matrix (in magnitude) learned by different GMM methods on the `Face` dataset with $m/n = 0.4$. It is preferred to view the figure electronically. The magnitudes in each precision matrix are scaled to the range of $[0, 1]$.

by the methods of PLE, Sparse-GMM, and Sparse-GMM(G) on the `face` dataset. From this figure, we can see that Sparse-GMM and Sparse-GMM(G) show much clearer groups sparsity than PLE, demonstrating the benifits of using group sparsity constructed from the banding patterns.

## 4.2 Real measurements

We demonstrate the efficacy of the proposed methods on the CS of video, with the measurements acquired by the actual hardware of CACTI camera [6]. A letter is placed on the blades of a chopper wheel that rotates at an angular velocity of 15 blades per second. The training data are obtained from the videos of a chopper wheel rotating at several orientations, positions and velocities. These training videos are captured by a regular camcorder at frame-rates that are different from the high-speed frame rate achieved by CACTI reconstruction. Other settings of the methods are the same as in the experi-

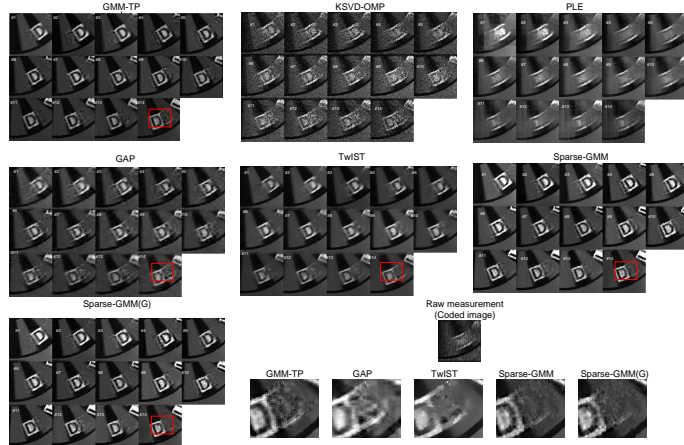

Figure 3: Reconstructed images $256 \times 256 \times T$ by different methods from the "raw measurement" acquired from CACTI with $T = 14$. The region in the red boxes are enlarged and shown at the right bottom part for better comparison.

ments on simulated data. The reconstruction results are shown in Figure 3, which shows that Sparse-GMM(G) generally yields sharper reconstructed frames with less ghost effects than other methods.

## 5 Conclusions

The success of compressive sensing of signals from a GMM highly depends on the quality of the estimator of the *unknown* GMM. In this paper, we have developed a hierarchical Bayesian method to simultaneously estimate the GMM and recover the signals, all based on using only incomplete linear measurements and a Bayesian shrinkage prior for promoting sparsity of the Gaussian precision matrices. In addition, we have obtained theoretical results under the challenging assumption that the underlying GMM is *unknown* and has to be estimated from measurements that contain only *incomplete* information about the signals. Our results extend substantially from previous theoretical results in [7] which assume the GMM is exactly known. The experimental results with both simulated and hardware-acquired measurements show the proposed method significantly outperforms state-of-the-art methods.

**Acknowledgement**
The research reported here was funded in part by ARO, DARPA, DOE, NGA and ONR.

## Footnotes

[1]A similar assumption is made in expression (3.13) of [22].

[2]The complete results can be found at the website: https://sites.google.com/site/nipssgmm/.

[3] It is downloaded from http://cs.nyu.edu/~roweis/data.html.

[4] The results of other settings can be found at https://sites.google.com/site/nipssgmm/.

[5] This subimage processing strategy has also been used in [2].

[6]The results of the training videos containing general scenes can be found at the aforementioned website.

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
