[Supplementary Material]

# Compressive Sensing of Signals from a GMM with Sparse Precision Matrices (Supplementary Material)

[1]**Jianbo Yang**   [1]**Xuejun Liao**   [2]**Minhua Chen**   [1] **Lawrence Carin**
[1]Department of Electrical and Computer Engineering, Duke University
[2]Department of Statistics & Department of Computer Science, University of Chicago
{jianbo.yang;xjliao;lcarin@duke@duke.edu},{dukemeeting@gmail.com}

## 1 Demonstration of GMRF on Image Signals

An image of size $321 \times 481$ randomly selected from the Berkeley Segmentation Dataset,[1] is partitioned into a collection of fully-overlapped equal-sized local patches, each of size $8 \times 8$, which are vectorized to constitute the image signals $\{\boldsymbol{x}_i\}_{i=1}^N$.[2] We use a standard GMM to fit the signals $\{\boldsymbol{x}_i\}_{i=1}^N$. The parameters are estimated by the EM algorithm [1], without any prior put on the parameters $\{\pi^{(k)}, \boldsymbol{\mu}^{(k)}, \boldsymbol{\Omega}^{(k)}\}$. The number of the GMM components is set as eight. Since $N$ is fairly large (more than 450,000) relative to the number of the parameters of the GMM (around 528), we mildly assume that this estimated GMM is a good approximation of the ground truth distribution $p(\boldsymbol{x})$ for the signals $\{\boldsymbol{x}_i\}_{i=1}^N$. Figure 1 plots the precision matrices of the eight Gaussian components of the estimated GMM.

It is evident from this figure that the precision matrices are sparse with many off-diagonal elements close to zero and each precision matrix has the banding pattern. The bands constituted by the significant nonzero elements are located in the main diagonal and a few lines below or above the main diagonal. As illustrated in Figure 2, those significant nonzero elements in the bands correspond to the pixel-pairs in which the two pixels are neighbors in a 2-D patch. Note that when one chooses other images, uses other patch sizes (e.g., $6 \times 6$, $12 \times 12$) or sets a different $K$ ($K$ could be tuned by Akaike information criterion or Bayesian information criterion, or inferred by a Dirichlet process), the resultant plots are similar to those in Figure 1 and 2.

## 2 Updates of variational EM algorithm

The full update expressions of the variational EM are given below.

**Update** $q(\mathbf{X}, \mathbf{Z})$: From

$$q^*(\mathbf{X}, \mathbf{Z}) \propto \exp\left(\mathbb{E}[\ln p(\mathbf{Y}|\mathbf{X}, \kappa)p(\mathbf{X}|\mathbf{Z}, \boldsymbol{\mu}, \boldsymbol{\Omega})p(\mathbf{Z}|\boldsymbol{\pi})]\right),$$

we obtain

$$q^*(\mathbf{X}, \mathbf{Z}) = \prod_{i=1}^N \rho_i^{(z_i)} \mathcal{N}(\boldsymbol{x}_i | \boldsymbol{\eta}_i^{(z_i)}, \mathbf{C}_i^{(z_i)})$$

Figure 1: The plots of the eight precision matrices of the GMM estimated from $\{x_i\}_{i=1}^{N}$ by an EM algorithm. The regions in the red dash-line boxes in the first and fourth plots are enlarged and shown on the top of the plots of the eight precision matrices. In the title of each plot, the index of the component as well as the corresponding weights of the mixture model are shown.

Figure 2: The diagram of the relationship between the 16 pixels in a $4 \times 4$ 2-D patch (right) and the 196 elements in a $16 \times 16$ precision matrix (left). The pairs of between a pixel (in yellow) and each of its four immediate neighbors (in blue), left, right, up and down, in the right figure are considered to have the conditional dependence relationship as highlighted in blue in the left figure. The region in the red dash-line box in the left figure shows the case of Pixel 7 and its four immediate neighbors.

with

$$\mathbf{C}_i^{(z_i)} = (\kappa^* \boldsymbol{\Phi}_i' \boldsymbol{\Phi}_i + \boldsymbol{\Omega}^{(z_i)^*})^{-1}$$

$$\boldsymbol{\eta}_i^{(z_i)} = \mathbf{C}_i^{(z_i)} (\kappa^* \boldsymbol{\Phi}_i' \mathbf{y}_i + \boldsymbol{\Omega}^{(z_i)^*} \boldsymbol{\mu}^{(z_i)^*})$$

$$\rho_i^{(z_i)} = \frac{\pi^{(z_i)} \mathcal{N}(\mathbf{y} | \boldsymbol{\Phi} \boldsymbol{\mu}^{(z_i)^*}, \kappa^{*-1} \mathbf{I} + \boldsymbol{\Phi} \boldsymbol{\Omega}^{(z_i)^{*-1}} \boldsymbol{\Phi}')}{\sum_{l=1}^{K} \pi^{(l)} \mathcal{N}(\mathbf{y} | \boldsymbol{\Phi} \boldsymbol{\mu}^{(l)^*}, \kappa^{*-1} \mathbf{I} + \boldsymbol{\Phi} \boldsymbol{\Omega}^{(l)^{*-1}} \boldsymbol{\Phi}')}.$$

Hence, $\mathbb{E}(\boldsymbol{x}_i) = \sum_{i=k}^{K} \rho_i^{(k)} \boldsymbol{\eta}_i^{(k)}$.

**Update** $q(\boldsymbol{\alpha})$: From

$$q^*(\boldsymbol{\alpha}) \propto \exp\left(\mathbb{E}[\ln p(\boldsymbol{\Omega}|\boldsymbol{\alpha})p(\boldsymbol{\alpha}|\boldsymbol{\gamma})]\right),$$

we obtain

$$q^*(\boldsymbol{\alpha}) = \prod_{k=1}^{K}\prod_{s=1}^{n}\prod_{t<s} \mathrm{InvGau}\left(\alpha_{ts}^{(k)}\Big| \sqrt{\frac{\mathbb{E}(\gamma_{ts}^{(k)})}{\omega_{ts}^{(k)*2}}}, \mathbb{E}(\gamma_{ts}^{(k)})\right),$$

where $\mathrm{InvGau}(x|g,h) = \sqrt{\frac{h}{2\pi x^3}}\exp(-\frac{h(x-g)^2}{2g^2 x})$ denotes an inverse Gaussian distribution. Hence,
$\mathbb{E}(\alpha_{ts}) = \sqrt{\frac{\mathbb{E}(\gamma_{ts}^{(k)})}{\omega_{ts}^{(k)*2}}}$ and $\mathbb{E}(\alpha_{ts}^{-1}) = (\sqrt{\frac{\mathbb{E}(\gamma_{ts}^{(k)})}{\omega_{ts}^{(k)*2}}})^{-1} + \mathbb{E}(\gamma_{ts}^{(k)})^{-1}$.

**Update** $q(\boldsymbol{\gamma})$: From

$$q^*(\boldsymbol{\gamma}) \propto \exp\left(\mathbb{E}[\ln p(\boldsymbol{\alpha}|\boldsymbol{\gamma})p(\boldsymbol{\gamma})]\right),$$

we obtain

$$q^*(\boldsymbol{\gamma}) = \prod_{k=1}^{K}\prod_{s=1}^{n}\prod_{t<s} \mathrm{Ga}\left(\gamma_{ts}^{(k)}\Big| a_0 + 1, b_0 + \frac{\mathbb{E}(\alpha_{ts}^{(k)^{-1}})}{2}\right).$$

Hence, $\mathbb{E}(\gamma_{ts}) = \frac{a_0 + 1}{b_0 + \frac{\mathbb{E}(\alpha_{ts}^{(k)^{-1}})}{2}}$.

**Update** $\kappa$:

$$\kappa^* = \arg\max_{\kappa}\left\{\mathbb{E}_{\mathbf{X}}[\ln p(\mathbf{Y}|\mathbf{X}, \kappa)p(\kappa)]\right\}$$

$$\kappa^* = \frac{\frac{Nm}{2} + e_0}{\frac{\sum_{i=1}^{N}\sum_{k=1}^{K} \rho_i^{(k)}[(\mathbf{y}_i - \boldsymbol{\Phi}_i\boldsymbol{\eta}_i^{(k)})'(\mathbf{y}_i - \boldsymbol{\Phi}_i\boldsymbol{\eta}_i^{(k)}) + \mathrm{tr}(\boldsymbol{\Phi}_i'\boldsymbol{\Phi}_i(\tilde{\boldsymbol{\Omega}}_i^{(k)})^{-1})]}{2} + f_0}$$

**Update** $\boldsymbol{\pi}$: Let $\eta^{(k)} = \alpha_0 + N^{(k)}$ with $N^{(k)} = \sum_{i=1}^{N} \rho_i^{(k)}$, then

$$\boldsymbol{\pi}^* = \arg\max_{\boldsymbol{\pi}}\left\{\mathbb{E}_{\mathbf{Z}}[\ln p(\mathbf{Z}|\boldsymbol{\pi})p(\boldsymbol{\pi})] + \lambda(\sum_{k=1}^{K}\pi^{(k)} - 1)\right\}$$

where $\lambda$ is a Lagrange multiplier. We obtain that

$$\pi^{(k)*} = \frac{\eta^{(k)} - 1}{\sum_{i=1}^{K}\eta^{(i)} - K}.$$

**Update** $\boldsymbol{\mu}^{(k)}$: Let $\bar{\boldsymbol{x}}^{(k)} = \frac{1}{N^{(k)}}\sum_{i=1}^{N}\rho_i^{(k)}\mathbb{E}(\boldsymbol{x}_i)$, then

$$\boldsymbol{\mu}^{(k)*} = \arg\max_{\boldsymbol{\mu}^{(k)}}\left\{\mathbb{E}_{\mathbf{X},\mathbf{Z}}[\ln p(\boldsymbol{\mu}^{(k)}|\mathbf{m}_0, \beta_0, \boldsymbol{\Omega}^{(k)})p(\mathbf{X}|\mathbf{Z}, \boldsymbol{\mu}, \boldsymbol{\Omega})]\right\}.$$

We obtain that

$$\boldsymbol{\mu}^{(k)*} = \frac{\beta_0\mathbf{m}_0 + N^{(k)}\bar{\boldsymbol{x}}^{(k)}}{\beta_0 + N^{(k)}}.$$

**Update** $\boldsymbol{\Omega}^{(k)}$:

$$\boldsymbol{\Omega}^{(k)*} = \arg\max_{\boldsymbol{\Omega}^{(k)}}\left\{\mathbb{E}_{\boldsymbol{\alpha},\mathbf{Z},\mathbf{X}}[\ln p(\boldsymbol{\Omega}^{(k)}|\boldsymbol{\alpha})p(\mathbf{X}|\mathbf{Z}, \boldsymbol{\mu}, \boldsymbol{\Omega})]\right\}$$

For convenience, we let

$$\boldsymbol{\Omega}^{(k)} = \left[ \begin{array}{cc} \boldsymbol{\Omega}_{\tilde{s}\tilde{s}}^{(k)} & \boldsymbol{\Omega}_{\tilde{s}s}^{(k)} \\ \boldsymbol{\Omega}_{s\tilde{s}}^{(k)} & \boldsymbol{\Omega}_{ss}^{(k)} \end{array} \right]$$

Using identity:

$$\mathcal{N}(\boldsymbol{x}_i|\boldsymbol{0}, \boldsymbol{\Omega}^{(k)^{-1}}) = \mathcal{N}(\boldsymbol{x}_{\tilde{s}i}| - (\boldsymbol{\Omega}_{\tilde{s}\tilde{s}}^{(k)})^{-1}\boldsymbol{\Omega}_{\tilde{s}s}^{(k)}\boldsymbol{x}_{si}, \boldsymbol{\Omega}_{\tilde{s}\tilde{s}}^{(k)^{-1}})\mathcal{N}(\boldsymbol{x}_{si}|0, (\boldsymbol{\Omega}_{ss}^{(k)} - \boldsymbol{\Omega}_{\tilde{s}s}^{(k)'}\boldsymbol{\Omega}_{\tilde{s}\tilde{s}}^{(k)^{-1}}\boldsymbol{\Omega}_{\tilde{s}s}^{(k)})^{-1})$$

To make $\boldsymbol{x}_i$ have the $\boldsymbol{0}$ mean, we let $\check{\boldsymbol{x}}_i = \boldsymbol{x}_i - \boldsymbol{\mu}^{(k)^*}$

$$(\boldsymbol{\Omega}_{\tilde{s}s}^{(k)^*}, \boldsymbol{\Omega}_{ss}^{(k)^*})$$

$$= \arg \max_{\boldsymbol{\Omega}_{ss}^{(k)}, \boldsymbol{\Omega}_{\tilde{s}s}^{(k)}} \mathbb{E}_{\mathbf{X}, \boldsymbol{\alpha}, \mathbf{Z}}[\log[\prod_{i=1}^N \mathcal{N}(\check{\boldsymbol{x}}_{\tilde{s}i}| - (\boldsymbol{\Omega}_{\tilde{s}\tilde{s}}^{(k)})^{-1}\boldsymbol{\Omega}_{\tilde{s}s}^{(k)}\check{\boldsymbol{x}}_{si}, \boldsymbol{\Omega}_{\tilde{s}\tilde{s}}^{(k)^{-1}})^{\mathbf{z}_{ik}}$$

$$\times \mathcal{N}(\check{\boldsymbol{x}}_{si}|0, (\boldsymbol{\Omega}_{ss}^{(k)} - \boldsymbol{\Omega}_{\tilde{s}s}^{(k)'}\boldsymbol{\Omega}_{\tilde{s}\tilde{s}}^{(k)^{-1}}\boldsymbol{\Omega}_{\tilde{s}s}^{(k)})^{-1})^{\mathbf{z}_{ik}} \times \mathcal{N}(\boldsymbol{\Omega}_{\tilde{s}s}^{(k)}|\boldsymbol{0}, \text{diag}^{-1}(\boldsymbol{\alpha}_{\tilde{s}s}^{(k)})) \times 1(\boldsymbol{\Omega}^{(k)} \in \boldsymbol{S}_+)]]$$

We can obtain

$$\boldsymbol{\Omega}_{\tilde{s}s}^{(k)^*} = (N^{(k)}\bar{\boldsymbol{\Sigma}}_{ss}^{(k)}(\boldsymbol{\Omega}_{\tilde{s}\tilde{s}}^{(k)^*})^{-1} + \text{diag}(\mathbb{E}(\boldsymbol{\alpha}_{\tilde{s}s}^{(k)})))^{-1}(-N^{(k)}\bar{\boldsymbol{\Sigma}}_{\tilde{s}s}^{(k)})$$

$$\boldsymbol{\Omega}_{ss}^{(k)^*} = ((\bar{\boldsymbol{\Sigma}}_{ss}^{(k)})^{-1} + \boldsymbol{\Omega}_{\tilde{s}s}^{(k)^{*'}}\boldsymbol{\Omega}_{\tilde{s}\tilde{s}}^{(k)^{*-1}}\boldsymbol{\Omega}_{\tilde{s}s}^{(k)^*})$$

where

$$\bar{\boldsymbol{\Sigma}}^{(k)} = \frac{1}{N^{(k)}} \sum_{i=1}^N \rho_i^{(k)}[(\boldsymbol{\eta}_i^{(k)} - \boldsymbol{\mu}^{(k)^*})(\boldsymbol{\eta}_i^{(k)} - \boldsymbol{\mu}^{(k)^*})' + \mathbf{C}_i^{(k)}].$$

The optimal $\boldsymbol{\Omega}^{(k)^*}$ computed by the above procedure is guaranteed to satisfied to the positive definite constraint, as proved in [2].

## 3 Proof of Theorem 1

*Proof of Theorem 1*: A Taylor expansion of the inverse matrix gives

$$\begin{aligned} &\left(\boldsymbol{\Omega}_0 + \boldsymbol{\Delta} + \boldsymbol{\Phi}_i'\boldsymbol{R}^{-1}\boldsymbol{\Phi}_i\right)^{-1} \\ =\ &\left[\mathbf{I} + \left(\boldsymbol{\Omega}_0 + \boldsymbol{\Phi}_i'\boldsymbol{R}^{-1}\boldsymbol{\Phi}_i\right)^{-1}\boldsymbol{\Delta}\right]^{-1}\left(\boldsymbol{\Omega}_0 + \boldsymbol{\Phi}_i'\boldsymbol{R}^{-1}\boldsymbol{\Phi}_i\right)^{-1}, \\ =\ &\sum_{n=0}^{\infty}(-1)^n\left[\left(\boldsymbol{\Omega}_0 + \boldsymbol{\Phi}_i'\boldsymbol{R}^{-1}\boldsymbol{\Phi}_i\right)^{-1}\boldsymbol{\Delta}\right]^n\left(\boldsymbol{\Omega}_0 + \boldsymbol{\Phi}_i'\boldsymbol{R}^{-1}\boldsymbol{\Phi}_i\right)^{-1}. \end{aligned} \quad (1)$$

Based on (1), we obtain the signal reconstruction error

$$\widetilde{\boldsymbol{x}}_i - \widehat{\boldsymbol{x}}_i = \sum_{n=1}^{\infty}\left[-\left(\boldsymbol{\Omega}_0 + \boldsymbol{\Phi}_i'\boldsymbol{R}^{-1}\boldsymbol{\Phi}_i\right)^{-1}\boldsymbol{\Delta}\right]^n\left(\boldsymbol{\Omega}_0 + \boldsymbol{\Phi}_i'\boldsymbol{R}^{-1}\boldsymbol{\Phi}_i\right)^{-1}\boldsymbol{\Phi}_i'\boldsymbol{R}^{-1}\left(\mathbf{y}_i - \boldsymbol{\Phi}_i\boldsymbol{\mu}\right).$$

Applying some norm, e.g., $\ell_2$ norm on vectors and $F$ norm on matrix, on both sides and using the properties of norms, we obtain

$$\begin{aligned} \|\widetilde{\boldsymbol{x}}_i - \widehat{\boldsymbol{x}}_i\|_2 &\le \sum_{n=1}^{\infty}\left\|\left(\boldsymbol{\Omega}_0 + \boldsymbol{\Phi}_i'\boldsymbol{R}^{-1}\boldsymbol{\Phi}_i\right)^{-1}\boldsymbol{\Delta}\right\|_F^n\left\|\left(\boldsymbol{\Omega}_0 + \boldsymbol{\Phi}_i'\boldsymbol{R}^{-1}\boldsymbol{\Phi}_i\right)^{-1}\boldsymbol{\Phi}_i'\boldsymbol{R}^{-1}\left(\mathbf{y}_i - \boldsymbol{\Phi}_i\boldsymbol{\mu}\right)\right\|_2, \\ &= \sum_{n=1}^{\infty}\left\|\left(\boldsymbol{\Omega}_0 + \boldsymbol{\Phi}_i'\boldsymbol{R}^{-1}\boldsymbol{\Phi}_i\right)^{-1}\boldsymbol{\Delta}\right\|_F^n\|\widehat{\boldsymbol{x}}_i - \boldsymbol{\mu}\|_2, \\ &= \frac{\left\|\left(\boldsymbol{\Omega}_0 + \boldsymbol{\Phi}_i'\boldsymbol{R}^{-1}\boldsymbol{\Phi}_i\right)^{-1}\boldsymbol{\Delta}\right\|_F}{1 - \left\|\left(\boldsymbol{\Omega}_0 + \boldsymbol{\Phi}_i'\boldsymbol{R}^{-1}\boldsymbol{\Phi}_i\right)^{-1}\boldsymbol{\Delta}\right\|_F}\|\widehat{\boldsymbol{x}}_i - \boldsymbol{\mu}\|_2, \end{aligned} \quad (2)$$

where the last equation arises from the assumption that $\left\| \left( \boldsymbol{\Omega}_0 + \boldsymbol{\Phi}_i' \boldsymbol{R}^{-1} \boldsymbol{\Phi}_i \right)^{-1} \boldsymbol{\Delta} \right\|_F < 1$.

Because $\left\| \left( \boldsymbol{\Omega}_0 + \boldsymbol{\Phi}_i' \boldsymbol{R}^{-1} \boldsymbol{\Phi}_i \right)^{-1} \boldsymbol{\Delta} \right\|_F \leq \left\| \left( \boldsymbol{\Omega}_0 + \boldsymbol{\Phi}_i' \boldsymbol{R}^{-1} \boldsymbol{\Phi}_i \right)^{-1} \right\|_F \| \boldsymbol{\Delta} \|_F$, one has

$$1 - \left\| \left( \boldsymbol{\Omega}_0 + \boldsymbol{\Phi}_i' \boldsymbol{R}^{-1} \boldsymbol{\Phi}_i \right)^{-1} \boldsymbol{\Delta} \right\|_F \geq 1 - \left\| \left( \boldsymbol{\Omega}_0 + \boldsymbol{\Phi}_i' \boldsymbol{R}^{-1} \boldsymbol{\Phi}_i \right)^{-1} \right\|_F \| \boldsymbol{\Delta} \|_F .$$

When $\left\| \left( \boldsymbol{\Omega}_0 + \boldsymbol{\Phi}_i' \boldsymbol{R}^{-1} \boldsymbol{\Phi}_i \right)^{-1} \right\|_F \| \boldsymbol{\Delta} \|_F < 1$, it follows from (2) that

$$\| \widetilde{\boldsymbol{x}}_i - \widehat{\boldsymbol{x}}_i \|_2 \quad \leq \quad \frac{\left\| \left( \boldsymbol{\Omega}_0 + \boldsymbol{\Phi}_i' \boldsymbol{R}^{-1} \boldsymbol{\Phi}_i \right)^{-1} \right\|_F \| \boldsymbol{\Delta} \|_F}{1 - \left\| \left( \boldsymbol{\Omega}_0 + \boldsymbol{\Phi}_i' \boldsymbol{R}^{-1} \boldsymbol{\Phi}_i \right)^{-1} \right\|_F \| \boldsymbol{\Delta} \|_F} \| \widehat{\boldsymbol{x}}_i - \boldsymbol{\mu} \|_2. \tag{3}$$

When $\boldsymbol{\mu} = \boldsymbol{0}$, we have

$$\| \widetilde{\boldsymbol{x}}_i - \widehat{\boldsymbol{x}}_i \|_2 \quad \leq \quad \frac{\left\| \left( \boldsymbol{\Omega}_0 + \boldsymbol{\Phi}_i' \boldsymbol{R}^{-1} \boldsymbol{\Phi}_i \right)^{-1} \right\|_F \| \boldsymbol{\Delta} \|_F}{1 - \left\| \left( \boldsymbol{\Omega}_0 + \boldsymbol{\Phi}_i' \boldsymbol{R}^{-1} \boldsymbol{\Phi}_i \right)^{-1} \right\|_F \| \boldsymbol{\Delta} \|_F} \| \widehat{\boldsymbol{x}}_i \|_2. \tag{4}$$

This completes the proof.

## 4  Proof of Theorem 2

*Proof of Theorem 2*: The main idea of proof is inspired by [3][4]. Recall that $\boldsymbol{\Omega}^* = \boldsymbol{\Omega}_0 + \boldsymbol{\Delta}$ is the estimate of $\boldsymbol{\Omega}_0$ that minimizes

$$\mathcal{F}(\boldsymbol{\Omega}) = \mathrm{tr}(\boldsymbol{\Omega} \boldsymbol{\Sigma}_{em}) - \log |\boldsymbol{\Omega}| + \sum_{(i,j) \in S \cup S^c} \eta_{ij} |\omega_{ij}| \tag{5}$$

where $\eta_{ij} = \frac{\sqrt{\tau \gamma_{ij}}}{N}$. Let $\boldsymbol{\Omega} = \boldsymbol{\Omega}_0 + \widehat{\boldsymbol{\Delta}}$, and consider the set

$$\Theta_N(M) = \left\{ \widehat{\boldsymbol{\Delta}} : \widehat{\boldsymbol{\Delta}} = \widehat{\boldsymbol{\Delta}}', \| \widehat{\boldsymbol{\Delta}} \|_F = M R_N \right\}$$

where

$$R_N = \sqrt{\frac{(n+s) \log n}{N}} + \sqrt{n + s} \bar{\sigma}_{de}.$$

Note that $\mathcal{F}(\boldsymbol{\Omega})$ is a convex function and $\mathcal{F}(\boldsymbol{\Omega}_0 + \boldsymbol{\Delta}) \leq \mathcal{F}(\boldsymbol{\Omega}_0)$. Therefore, if we prove that the following probability tends to $1$[3]

$$p(\inf\{\mathcal{F}(\boldsymbol{\Omega}_0 + \widehat{\boldsymbol{\Delta}}) | \widehat{\boldsymbol{\Delta}} \in \Theta_N(M)\} > \mathcal{F}(\boldsymbol{\Omega}_0)). \tag{6}$$

then $\boldsymbol{\Delta}$ corresponding to the minimizer $\boldsymbol{\Omega}^*$ are inside the sphere defined by $\Theta_N(M)$ and $\| \boldsymbol{\Delta} \|_F \leq M R_N$ with high probability, equivalently, $\| \boldsymbol{\Omega}^* - \boldsymbol{\Omega}_0 \|_F^2 = \mathcal{O}_p\{\sqrt{(n+s) \log n / N} + \sqrt{n + s} \bar{\sigma}_{de}\}$. In the following, we give the proof of expression (6).

According to (5), we obtain that

$$\mathcal{F}(\boldsymbol{\Omega}) - \mathcal{F}(\boldsymbol{\Omega}_0) = I_1 + I_2 + I_3,$$

where

$$I_1 = \mathrm{tr}(\boldsymbol{\Omega} \boldsymbol{\Sigma}_{em}) - \mathrm{tr}(\boldsymbol{\Omega}_0 \boldsymbol{\Sigma}_{em}) - \mathrm{tr}(\widehat{\boldsymbol{\Delta}} \boldsymbol{\Sigma}_0) = \mathrm{tr}(\widehat{\boldsymbol{\Delta}} (\boldsymbol{\Sigma}_{em} - \boldsymbol{\Sigma}_0))$$

$$I_2 = - \log |\boldsymbol{\Omega}| + \log |\boldsymbol{\Omega}_0| + \mathrm{tr}(\widehat{\boldsymbol{\Delta}} \boldsymbol{\Sigma}_0) = \mathrm{vec}(\widehat{\boldsymbol{\Delta}})' \left[ \int_0^1 (1 - \nu)(\boldsymbol{\Omega}_0 + \nu \widehat{\boldsymbol{\Delta}})^{-1} \otimes (\boldsymbol{\Omega}_0 + \nu \widehat{\boldsymbol{\Delta}})^{-1} \mathrm{d}\nu \right] \mathrm{vec}(\widehat{\boldsymbol{\Delta}})$$

$$I_3 = \sum_{(i,j) \in S \cup S^c} \eta_{ij} (|\omega_{ij}| - |\omega_{ij}^0|)$$

where $I_2$ is obtained by using Taylor's expansion, and $\text{vec}(\widehat{\boldsymbol{\Delta}})$ denotes the vectorized $\widehat{\boldsymbol{\Delta}}$ and $\otimes$ represents the Kronecker product.

**For $I_1$:**

Let $\Delta_{ij}$, $\sigma_{ij}^{em}$, $\sigma_{ij}^{em0}$, $\sigma_{ij}^{de}$, $\sigma_{ij}^{0}$ are the elements at the $i$-th row and $j$-th column of $\widehat{\boldsymbol{\Delta}}$, $\boldsymbol{\Sigma}_{em}$, $\boldsymbol{\Sigma}_{em}^{0}$, $\boldsymbol{\Sigma}_{de}$, $\boldsymbol{\Sigma}_0$, respectively. We obtain the following inequality for $I_1$ as

$$I_1 \geq -|I_1| \geq -\underbrace{|\sum_{i \neq j}(\sigma_{ij}^{em} - \sigma_{ij}^0)\widehat{\Delta}_{ij}|}_{I_1^{(1)}} - \underbrace{|\sum_{i=1}^{n}(\sigma_{ii}^{em} - \sigma_{ii}^0)\widehat{\Delta}_{ii}|}_{I_1^{(2)}}.$$

Based on the Lemma 1 of [4] and Lemma 2 of [3], we obtain that, with probability tending to 1,

$$\max_{i \neq j}|\sigma_{ij}^{em} - \sigma_{ij}^0| \leq \max_{i \neq j}\{|\sigma_{ij}^{em0} - \sigma_{ij}^0| + |\sigma_{ij}^{de}|\}$$

$$\leq C_1\sqrt{\frac{\log n}{N}} + \bar{\sigma}_{de}$$

$$\leq C_1\left(\sqrt{\frac{\log n}{N}} + \bar{\sigma}_{de}\right).$$

where $C_1 \geq 1$. Let $\widehat{\boldsymbol{\Delta}}^{+}$ be a diagonal matrix with the diagonal same as $\widehat{\boldsymbol{\Delta}}$ and $\widehat{\boldsymbol{\Delta}}^{-} = \widehat{\boldsymbol{\Delta}} - \widehat{\boldsymbol{\Delta}}^{+}$. We obtain

$$I_1^{(1)} \leq C_1\left(\sqrt{\frac{\log n}{N}} + \bar{\sigma}_{de}\right)|\widehat{\boldsymbol{\Delta}}^{-}|_1$$

where $|\mathbf{X}|$ is the $\ell_1$ norm of the vectorized matrix $\mathbf{X}$.

Similarly, according to Cauchy-Schwartz inequality, we obtain the expression

$$I_1^{(2)} \leq \left[\sum_{i=1}^{n}(\sigma_{ii}^{em} - \sigma_{ii}^0)^2\right]^{1/2}\|\widehat{\boldsymbol{\Delta}}^{+}\|_F$$

$$\leq \sqrt{n}\max_{1 \leq i \leq n}|\sigma_{ii}^{em} - \sigma_{ii}^0|\|\widehat{\boldsymbol{\Delta}}^{+}\|_F$$

$$\leq \sqrt{n}\max_{1 \leq i \leq n}(|\sigma_{ii}^{em0} - \sigma_{ii}^0| + |\sigma_{ii}^{de}|)\|\widehat{\boldsymbol{\Delta}}^{+}\|_F$$

$$\leq \left(C_2\sqrt{\frac{n\log n}{N}} + \sqrt{n}\bar{\sigma}_{de}\right)\|\widehat{\boldsymbol{\Delta}}^{+}\|_F$$

$$\leq C_2\left(\sqrt{\frac{(n+s)\log n}{N}} + \sqrt{n+s}\bar{\sigma}_{de}\right)\|\widehat{\boldsymbol{\Delta}}^{+}\|_F.$$

holds with also probability tending to 1, where $C_2 \geq 1$.

**For $I_2$**

Let $\|\mathbf{X}\|$ be the operator norm defined by $\|\mathbf{X}\| = \sqrt{\psi_{\max}(\mathbf{X}\mathbf{X}')}$. We obtain that

$$I_2 \geq \int_0^1 (1-\nu)\min_{0 \leq \nu \leq 1}\psi_{\min}[(\boldsymbol{\Omega}_0 + \nu\widehat{\boldsymbol{\Delta}})^{-1} \otimes (\boldsymbol{\Omega}_0 + \nu\widehat{\boldsymbol{\Delta}})^{-1}]\mathrm{d}\nu\|\widehat{\boldsymbol{\Delta}}\|_F^2$$

$$\geq \frac{1}{2}\|\widehat{\boldsymbol{\Delta}}\|_F^2 \min_{0 \leq \nu \leq 1}\psi_{\max}^{-2}(\boldsymbol{\Omega}_0 + \nu\widehat{\boldsymbol{\Delta}})$$

$$\geq \frac{1}{2}\|\widehat{\boldsymbol{\Delta}}\|_F^2(\|\boldsymbol{\Omega}_0\| + \|\widehat{\boldsymbol{\Delta}}\|)^{-2}$$

$$\geq \frac{1}{2}\|\widehat{\boldsymbol{\Delta}}\|_F^2(\tau_1^{-1} + o(1))^{-2}$$

with probability tending to 1, since $\|\widehat{\boldsymbol{\Delta}}\| \leq \|\widehat{\boldsymbol{\Delta}}\|_F = o(1)$, where the symbols $\psi_{\min}(\mathbf{X})$ and $\psi_{\max}(\mathbf{X})$ denote the smallest and largest eigenvalues of matrix $\mathbf{X}$ respectively.

**For $I_3$**

Since

$$
|\omega_{ij}| - |\omega_{ij}^0| = \left\{ \begin{array}{ll} |\omega_{ij}| = |\widehat{\Delta}_{ij}| & \text{if } (i,j) \in S^c \\ |\omega_{ij} + \widehat{\Delta}_{ij}| - |\omega_{ij}^0| \geq -|\widehat{\Delta}_{ij}| & \text{if } (i,j) \in S \end{array} \right. ,
$$

we obtain that

$$
I_3 \geq \sum_{(i,j)\in S^c} \eta_{ij}|\widehat{\Delta}_{ij}| - \sum_{(i,j)\in S} \eta_{ij}|\widehat{\Delta}_{ij}| \geq \underline{\eta}|\widehat{\boldsymbol{\Delta}}_{S^c}^-| - \overline{\eta}|\widehat{\boldsymbol{\Delta}}_S^-|.
$$

Let

$$
\underline{\eta} = \frac{C_1}{\epsilon_1}\left(\sqrt{\frac{\log n}{N}} + \bar{\sigma}_{de}\right)
$$

$$
\overline{\eta} = \frac{C_1}{\epsilon_2}\left(\sqrt{\frac{\log n}{N}} + \bar{\sigma}_{de}\right).
$$

We obtain that

$$
\begin{aligned}
& I_1 + I_2 + I_3 \\
\geq & \frac{1}{2}\|\widehat{\boldsymbol{\Delta}}\|_F^2(\tau_1^{-1} + o(1))^{-2} - C_1\left(\sqrt{\frac{\log n}{N}} + \bar{\sigma}_{de}\right)|\widehat{\boldsymbol{\Delta}}^-|_1 \\
& - C_2\left(\sqrt{\frac{(n+s)\log n}{N}} + \sqrt{n+s}\bar{\sigma}_{de}\right)\|\widehat{\boldsymbol{\Delta}}^+\|_F + \underline{\eta}|\widehat{\boldsymbol{\Delta}}_{S^c}^-| - \overline{\eta}|\widehat{\boldsymbol{\Delta}}_S^-| \\
\geq & \frac{1}{2}\|\widehat{\boldsymbol{\Delta}}\|_F^2(\tau_1^{-1} + o(1))^{-2} - C_1\left(\sqrt{\frac{\log n}{N}} + \bar{\sigma}_{de}\right)(1 - \frac{1}{\epsilon_1})|\widehat{\boldsymbol{\Delta}}_{S^c}^-|_1 \\
& - C_1\left(\sqrt{\frac{\log n}{N}} + \bar{\sigma}_{de}\right)(1 + \frac{1}{\epsilon_2})|\widehat{\boldsymbol{\Delta}}_S^-|_1 - C_2\left(\sqrt{\frac{(n+s)\log n}{N}} + \sqrt{n+s}\bar{\sigma}_{de}\right)\|\widehat{\boldsymbol{\Delta}}^+\|_F \\
\geq & \frac{1}{2}\|\widehat{\boldsymbol{\Delta}}\|_F^2(\tau_1^{-1} + o(1))^{-2} - C_1\left(\sqrt{\frac{\log n}{N}} + \bar{\sigma}_{de}\right)(1 + \frac{1}{\epsilon_2})\sqrt{s+n}\|\widehat{\boldsymbol{\Delta}}\|_F \\
& - C_2\left(\sqrt{\frac{(n+s)\log n}{N}} + \sqrt{n+s}\bar{\sigma}_{de}\right)\|\widehat{\boldsymbol{\Delta}}\|_F \\
\geq & \|\widehat{\boldsymbol{\Delta}}\|_F^2\left(\frac{1}{2}(\tau_1^{-1} + o(1))^{-2} - \frac{C_1(1+\epsilon_2)}{\epsilon_2 M} - \frac{C_2}{M}\right) \geq 0
\end{aligned}
$$

for sufficiently large $M$. The third inequality is hold, since the term $-C_1\left(\sqrt{\frac{\log n}{N}} + \bar{\sigma}_{de}\right)(1 - \frac{1}{\epsilon_1})|\widehat{\boldsymbol{\Delta}}_{S^c}^-|_1$ is always positive as similarly set in [4] and also

$$
|\widehat{\boldsymbol{\Delta}}_S^-|_1 \leq \sqrt{s}\|\widehat{\boldsymbol{\Delta}}_S^-\|_F \leq \sqrt{s}\|\widehat{\boldsymbol{\Delta}}^-\|_F \leq \sqrt{s+n}\|\widehat{\boldsymbol{\Delta}}^-\|_F.
$$

This completes the proof.

## 5 Proof of Theorem 3

*Proof*: According to Theorem 1 and Theorem 2, we can find an appropriate constant $M$ make the inequality $\|\mathbf{\Delta}\|_F \leq MC + M\sqrt{n+s}\bar{\sigma}_{de}$ hold with high probability, and further obtain that

$$
\begin{aligned}
\frac{1}{N}\sum_{i=1}^{N}\|\boldsymbol{\delta}_i\|_2 &= \frac{1}{N}\sum_{i=1}^{N}\|\widetilde{\boldsymbol{x}}_i - \widehat{\boldsymbol{x}}_i\|_2\\
&\leq \frac{1}{N}\sum_{i=1}^{N}\left\{\frac{\left\|\left(\mathbf{\Omega}_0 + \mathbf{\Phi}_i'\boldsymbol{R}^{-1}\mathbf{\Phi}_i\right)^{-1}\right\|_F \|\mathbf{\Delta}\|_F}{1 - \left\|\left(\mathbf{\Omega}_0 + \mathbf{\Phi}_i'\boldsymbol{R}^{-1}\mathbf{\Phi}_i\right)^{-1}\right\|_F \|\mathbf{\Delta}\|_F}\|\widehat{\boldsymbol{x}}_i - \boldsymbol{\mu}\|_2\right\}\\
&\leq \frac{\max_i\left\|\left(\mathbf{\Omega}_0 + \mathbf{\Phi}_i'\boldsymbol{R}^{-1}\mathbf{\Phi}_i\right)^{-1}\right\|_F \|\mathbf{\Delta}\|_F}{1 - \max_i\left\|\left(\mathbf{\Omega}_0 + \mathbf{\Phi}_i'\boldsymbol{R}^{-1}\mathbf{\Phi}_i\right)^{-1}\right\|_F \|\mathbf{\Delta}\|_F}\left\{\frac{1}{N}\sum_{i=1}^{N}\|\widehat{\boldsymbol{x}}_i - \boldsymbol{\mu}\|_2\right\}\\
&\leq \frac{\xi\left(MC + MD\bar{\sigma}_{de}\right)}{N(1 - \zeta)}\sum_{i=1}^{N}\|\widehat{\boldsymbol{x}}_i - \boldsymbol{\mu}\|_2 \qquad (7)\\
&\leq \frac{\xi\left(MC + \frac{MD}{N}\sum_{i=1}^{N}(2\widehat{\boldsymbol{x}}_{\max}\|\boldsymbol{\delta}_i\|_2 + \|\boldsymbol{\delta}_i\|_2^2 + \mathrm{tr}(\mathbf{C}_i))\right)}{N(1 - \zeta)}\sum_{i=1}^{N}\|\widehat{\boldsymbol{x}}_i - \boldsymbol{\mu}\|_2\\
&\leq \frac{\xi}{N(1 - \zeta)}\left(MC + \frac{MD}{N}\sum_{i=1}^{N}(2\widehat{\boldsymbol{x}}_{\max}\|\boldsymbol{\delta}_i\|_2 + \|\boldsymbol{\delta}_i\|_2^2 + \mathrm{tr}(\mathbf{C}_i))\right)\sum_{i=1}^{N}\|\widehat{\boldsymbol{x}}_i - \boldsymbol{\mu}\|_2\\
&\leq \frac{\xi}{N(1 - \zeta)}\left(MC + \frac{MD}{N}\sum_{i=1}^{N}(2\widehat{\boldsymbol{x}}_{\max}\|\boldsymbol{\delta}_i\|_2 + \delta_{\max}\|\boldsymbol{\delta}_i\|_2 + \mathrm{tr}(\mathbf{C}_i))\right)\sum_{i=1}^{N}\|\widehat{\boldsymbol{x}}_i - \boldsymbol{\mu}\|_2.
\end{aligned}
$$

where $C = \sqrt{(n+s)\log n/N}$, $D = \sqrt{n+s}$. The fourth inequality holds because

$$
\begin{aligned}
\bar{\sigma}_{de} &= \|\mathbf{\Sigma}_{de}\|_{\max} \leq \mathrm{tr}(\mathbf{\Sigma}_{de})\\
&\leq \frac{2}{N}\sum_{i=1}^{N}\|\widehat{\boldsymbol{x}}_i\|_2\|\boldsymbol{\delta}_i\|_2 + \mathrm{tr}\left(\frac{1}{N}\sum_{i=1}^{N}(\boldsymbol{\delta}_i\boldsymbol{\delta}_i' + \mathbf{C}_i)\right)\\
&= \frac{2}{N}\sum_{i=1}^{N}\|\widehat{\boldsymbol{x}}_i\|_2\|\boldsymbol{\delta}_i\|_2 + \frac{1}{N}\sum_{i=1}^{N}(\|\boldsymbol{\delta}_i\|_2^2 + \mathrm{tr}(\mathbf{C}_i)) \qquad (8)\\
&\leq \frac{1}{N}\sum_{i=1}^{N}(2\widehat{\boldsymbol{x}}_{\max}\|\boldsymbol{\delta}_i\|_2 + \|\boldsymbol{\delta}_i\|_2^2 + \mathrm{tr}(\mathbf{C}_i))
\end{aligned}
$$

where the second inequality holds based on Cauchy–Schwarz inequality.

Letting $\epsilon = \frac{1}{N}\sum_{i=1}^{N}\|\boldsymbol{\delta}_i\|_2$, we write from (7) that

$$
\frac{1 - \zeta}{\xi\epsilon_0 M}\epsilon - C \leq D((\delta_{\max} + 2\widehat{\boldsymbol{x}}_{\max})\epsilon + \upsilon).
$$

When $\frac{1-\zeta}{\epsilon_0 MD} > \xi(\delta_{\max} + 2\widehat{\boldsymbol{x}}_{\max})$, one has

$$
\begin{aligned}
\epsilon &\leq \frac{C/D + \upsilon}{\frac{1-\zeta}{\xi\epsilon_0 MD} - (\delta_{\max} + 2\widehat{\boldsymbol{x}}_{\max})},\\
&= \frac{\sqrt{(\log n)/N} + \upsilon}{\frac{1-\zeta}{\xi\epsilon_0 M\sqrt{n+s}} - (\delta_{\max} + 2\widehat{\boldsymbol{x}}_{\max})}\\
&= \frac{\sqrt{(\log n)/N} + \upsilon}{(1 - \zeta)/\sqrt{n+s} - M\epsilon_0(\delta_{\max} + 2\widehat{\boldsymbol{x}}_{\max})\xi}M\epsilon_0\xi.
\end{aligned}
$$

This completes the proof.

## Footnotes

[1]http://www.eecs.berkeley.edu/Research/Projects/CS/vision/grouping/resources.html

[2]These image signals can be taken as the groundtruth when one performs CS on the image signal $\{\boldsymbol{x}_i\}_{i=1}^N$.

[3]More exactly, the claim hold with probability greater than $1 - c_1 \exp(-c_2 \log n)$, where $c_1, c_2 > 0$ are independent of other parameters.