[Reviews · NeurIPS 2014]

Submitted by Assigned_Reviewer_4

This paper proposes a method to recover signals from compressive measurements. The method consists of jointly estimating the signal and a Gaussian Mixture Model (GMM) capable of representing it succinctly. The main contribution of the paper is the idea of imposing a sparse structure on the GMM adapted to the case when the signal of interest corresponds to image patches. This is further exploited by a more structured prior that promotes an appropriate group-sparsity pattern (essentially interactions between adjoining pixels are not penalized by the sparsity-inducing penalty). Some theoretical results are provided for the case in which there is just one Gaussian in the mixture, but their implication or the conditions under which they hold are not explained clearly. Numerical simulations show that the method yields good results for very small images and for individual image patches. However the question of how to apply the method to normal-sized images is not addressed.

This paper presents an interesting and potentially useful method for the compressed sensing of image patches and provides some hopeful numerical results. Its main weaknesses are:

-The theoretical results are very confusing. It is very difficult to derive any conclusions from the statement of the theorems or the additional explanations, which are very vague. The conditions for the theorems, which seem quite strong and ad hoc, are not explained. For example, a bound delta_max on the error seems to be assumed in Theorem 3. Also, the results in Theorem 2 are asymptotic and in high probability but this is not mentioned anywhere (except in the proof within the supplementary material). Finally, no attempt is made to clarify for what kind of sensing matrices the results would hold.

-The extension of the theory to the case of several Gaussians, which does not seem trivial and is the method that is actually applied in practice, is not discussed.

-Not a lot of visual results are shown either in the paper or in the supplementary material (this would be a good sanity check and I encourage the authors to include them).

-It is not discussed how to combine the patches obtained from the method to produce whole images. Reference [3] states that this is not trivial, so the authors should at least discuss it.

Quality:

The paper introduces a novel approach to signal estimation using GMMs by promoting sparsity and group sparsity in the covariance matrices. The authors provide some numerical evidence for very small images and image patches, although they do not tackle the reconstruction of whole images. The theoretical results are not very conclusive.

Clarity:

The paper is relatively clearly written, except for the more technical parts which are confusing and difficult to follow. This is apparent in Section 3, where the authors barely discuss the conditions and the results of the theorems. Theorem 3 for example can only hold with high probability and asymptotically as it is “a combination of Theorems 1 and 2”, but this is not even mentioned!

Some of the other mathematical statements are not very rigorous. For example, in equation (15) it seems that the authors consider x hat and x tilde random variables but they do not clarify in what sense the equality holds. The equality “follows from [21]” if delta_i is independent of x_i, but it is not clear to me that this is true. If it is, the authors should explain in what sense (in expectation, with high probability, etc.). Also, the authors refer to other works: “Following [21]” (line 280), “Similar to [21]” (line 284) … in a very vague way without specifying what part of the cited works they mean, which makes it difficult for the reader to follow.

Originality:

The idea of promoting a sparse or group-sparse structure of the covariance matrices in a GMM while performing signal estimation is novel to the best of my knowledge.

Significance:

I don’t think that the contribution of the paper is very significant because of the limited scope of the theory and because the numerical results do not include reconstructions of whole images.

After reading the authors' rebuttal, my views on the paper are basically unchanged, although I'm increasing the score to 6. In particular, even though it might be straightforward to combine the patches, this does not mean that they should produce an accurate estimate of features in the image that span several patches. However, it is perhaps understandable that the authors do not address this given the space constraints.
Summary: The paper introduces a novel approach to signal estimation using GMMs by promoting sparsity and group sparsity in the covariance matrices. The authors provide some numerical evidence for small images and image patches, although they do not tackle the reconstruction of whole images, and some theoretical results that are not very conclusive.

Submitted by Assigned_Reviewer_28

The paper presents an algorithm for recovering signals from its compressed measurements. The signals are assumed to be generated from a Gaussian mixture model (GMM), with each mixture component having a sparse precision matrix. Both the signals and the GMM parameters are estimated from the compressed samples. A hierarchical Bayesian approach is used for modeling the GMM and the inference is performed using variational techniques. The paper also presents theoretical guarantees for signal recovery and GMM model estimation.

Overall the paper is well written and there is some novelty in the approach. The main motivation for the paper is that instead of first learning GMM prior from a separate training dataset and then performing signal recovery, we should simultaneously estimate the GMM parameters and the signal from its compressed measurements. This can mitigate the effect of training-testing mismatch. However, the proposed hierarchical Bayesian inference seems computationally expensive and the paper do not mention anything about the computational complexity or time. Given the computational complexity, one might raise the question about the practicality of this approach in solving real-world problems. The experimental section could be improved further, see detailed comments below.

Novelty: The paper is well written and the approach seems novel to me.

Quality of references: Seem sufficient.

Reproducibility: The variational EM inference algorithm seems quite complicated. The authors should consider releasing their code.

I have one concern/question regarding the sparse precision matrix assumption. Can this model handle edges and high-frequency structures ?

The banding structure is odd as well. This would automatically penalize against edge structure --- would it not ?
Especially for temporal data that are highly oriented, it might be better NOT to enforce any additional structure --- but that would come with increased requirements on training data.

I have following concerns in the experimental section
1. In compressive sensing of images, the images has been resized to 12*12, which means it has lost all the details, such as edges. It doesn’t make much sense in reconstructing such signals.
2. Given that the images are just 12*12, one could easily learn GMM prior using natural images as training data. I suggest that the authors also show results for GMM-TP.
3. For compressive sensing of high speed video, is the patch/block size 64*64*8? For such as big patch size, many more mixture components should be learnt for GMM-TP. Also instead of learning GMM-TP using only traffic videos (which are very different from test video), it should be learnt using general motion videos.
Summary: A well motivated paper with a good mix of theoretical and experimental results. My concerns are with the models ability to capture high-frequency details as well as some strange choices in the experiments.

Submitted by Assigned_Reviewer_41

The authors introduce hierarchical Bayesian model for simultaneously estimating the parameters of a GMM with sparse precision matrices, along with recovering signals from incomplete linear measurements.

The sparsity is enforced through a Laplacian shrinkage prior on the elements of each precision, coupled with adaptive coefficients (gamma) to each element. (Akin to adaptive Bayesian graphical LASSO.) Additionaly, the GMM parameters also have appropriate priors. Two sparsity schemes are proposed, both with superior performance to state-of-the-art approaches: one with standard element-wise sparsity on the precision, and one with group-sparsity that exploits the spatial structure inherent in image and video signals (pixel/frame neighborhoods).

Optimization is done in a standard fashion, through variational-Bayes expectation-maximisation, to deal with intractabilities of the true posterior on each precision.

It is nice that the authors also present a few theoretical guarantees on the estimated precisions, that extend prior work to the case of an unknown underlying GMM.

The superiority of Sparse-GMM and Sparse-GMM(G) is validated through a few sparse signal recovery tasks on small images and video, with convincing results. Thought the conclusions from the video example (Figure 3) are rather qualitative, and could be argued that superiority is also shared by GMM-TP.

I believe the model, optimisation and bounds make for a nice result that could be easily added to the recent stream of excitement for GMMs as models of sparse image and video signals.

The quality and clarity of the paper are good. The ideas seem quite original with a strong potential for impact - maintly due to the simultaneous estimation scheme - widely extending the applicability of GMM models (though I am not an expert in this area).

Other comments:

Abstract: Though you do explain these in the main body, referring to 'partially observed' and 'incomplete linear measumerents' in the abstract might go over the reader's head. You repeat these terms in the intro prior to any explanation.

line 53: I suggest rephrasing 'in situ'.

line 219: Define Z.

line 330: Define 'peak signal-to-noise-ratio'.

Figure 2 caption: matrices -> matrix
Summary: A novel Bayesian hierarchical model for simultaneously estimating the parameters of a GMM with sparse precision matrices, along with recovering signals from incomplete linear measurements. Optimization is done in a standard fashion and new estimation guarantees are presented that apply to cases with unknown GMM parameters. I believe this widely extends the applicability of GMM models for sparse signals.
Author Feedback
Author rebuttal: We thank the reviewers for their valuable comments. We provide our feedback below, and the supporting experimental results are provided at the Support Site https://sites.google.com/site/nipssgmm/

Based on all reviewers’ comments, we have also revised the paper and the supplementary, and all the changes are highlighted in red. The revised paper and supplementary are also provided at Support Site at
https://sites.google.com/site/nipssgmm/paper_and_supp

Reviewer 1:

The banding pattern observed in the precision matrix (not the covariance matrix) is the property of the conditional independence rather than correlations/similarity, and it is applicable to all the cases where Gaussian MRF can be applied, including images with edge structure.

We have noticed that temporal data are highly oriented. Therefore, in the original submission, for video data, we don't add the banding pattern structure on the temporal domain but only on the spatial domain, i.e., $L_1 = {(i,j): pixel i is one of four immediate neighbors of pixels j in spatial domain, i\ne j}$. This can be verified in Fig. 2 of the paper. The patch size is 4X4X8 in every block 64X64X8 in the video data.

The experiments on USPS and Face images are only for illustration's purpose. The results on the USPS data with the original size 16X16 and the results of the face data with the different settings of the patch sizes can be found at the Support Site. We found that the proposed methods consistently perform better than other methods.

Following your suggestion, we have used additional general motion video to learn the GMM-TP in the video CS experiment, and the results are provided at the Support Site. It is seen that the performance of GMM-TP does not change much due to these additional training video, and the ranking of the methods is not changed in terms of PSNR.

Reviewer 2:

We respectfully disagree with the reviewer that the proposed method works only for "very small images and for individual image patches" and "the question of how to apply the method to normal-sized images is not addressed". In the experiments on CS of high-speed video, we have considered video of normal sizes 256X256X32 and 256X256X14 (see the 2nd part of Sec. 4.1 and Sec. 4.2), and the CS data are collected by real camera hardware (CACTI). Note that the CACTI hardware allows us to reconstruct video frames patch by patch. Combining the patches into a whole image is a standard procedure, and it can be achieved by the Matlab command "col2im". We will release the source code to the public upon acceptance of the paper.

We have provided additional visual results at the Support Site, where it is seen that the visual results are consistent with the PSNR results for the simulation datasets.

Regarding the theory, we have used the conclusion of Theorem 2 as a condition of Theorem 3, which makes Theorem 3 true without a probability. In the revised paper, we have re-stated Theorem 3 by replacing (in the premise part) the conclusion of Theorem 2 with the condition of Theorem 2, which makes the result of Theorem 3 hold with only with a probability.

The assumption that $delta_i$ is independent of $x_i$ is mainly motivated by [21, the 3rd paragraph of Sec. 3.3] and simplification. In this revised paper, we have removed this assumption, leading to a new definition of \Sigma_{de} but unchanged statement of Theorem 2, a mild change of the bound but unchanged asymptotic conclusions (see the last paragraph of Section 3.1) in Theorem 3.

We agree that the conditions of Theorem 3 are a little messy. However, we believe that this theorem is a useful first step towards the challenging goal of extending the theory of [7, Renna et. al, 2014] to the case when the Gaussian signal model is unknown and must be estimated along with the signals.

Reviewer 3:

Thank you for pointing out that our theoretic analysis " extend prior work to the case of an unknown underlying GMM."

We will follow your comments to improve the abstract and introduction in the final version.